# Plant defense phenotypes determine the consequences of volatile emission for individuals and neighbors

**Meredith C Schuman[1]\*, Silke Allmann[1,2], Ian T Baldwin[1]**

[1]Department of Molecular Ecology, Max Planck Institute for Chemical Ecology, Jena, Germany; [2]Department of Plant Physiology, Swammerdam Institute of Life Sciences, University of Amsterdam, Amsterdam, Netherlands

**Abstract** Plants are at the trophic base of terrestrial ecosystems, and the diversity of plant species in an ecosystem is a principle determinant of community structure. This may arise from diverse functional traits among species. In fact, genetic diversity within species can have similarly large effects. However, studies of intraspecific genetic diversity have used genotypes varying in several complex traits, obscuring the specific phenotypic variation responsible for community-level effects. Using lines of the wild tobacco *Nicotiana attenuata* genetically altered in specific well-characterized defense traits and planted into experimental populations in their native habitat, we investigated community-level effects of trait diversity in populations of otherwise isogenic plants. We conclude that the frequency of defense traits in a population can determine the outcomes of these traits for individuals. Furthermore, our results suggest that some ecosystem-level services afforded by genetically diverse plant populations could be recaptured in intensive monocultures engineered to be functionally diverse.

\*For correspondence:
mschuman@ice.mpg.de

## Introduction

In *The Origin of Species*, Darwin wrote that 'varieties are species in the process of formation' (*Darwin, 1876*). We now know that Darwin's 'varieties' are polymorphic characters resulting from natural genetic variation within species. The geneticist *Dobzhansky (1973)* echoed Darwin's view: 'It is evident that the differences among proteins at the levels of species, genus, family, order, class, and phylum are compounded of elements that vary also among individuals within a species'. It has also long been known that intraspecific genetic variation in plants can alter ecological community structure; for example, in 1960, De Wit reported that the reproductive output in mixtures of the grass species *Anthoxanthum odoratum* and *Phleum pratens* depended on associations of particular genotypes (*De Wit, 1960*; *Allard and Adams, 1969*). However, efforts to measure the broader consequences for communities are more recent (*Antonovics, 1992*; reviewed in *Haloin and Strauss, 2008*) and include the recently described field of community genetics.

Community genetics aims to quantify how genetic variation in specific traits affects the structure of ecological communities (reviewed in *Whitham et al., 2006*; *Haloin and Strauss, 2008*; *Whitham et al., 2012*). Genes most likely to have community effects are those with a large effect in the foundation species that structure communities (*Whitham et al., 2006*), and plants are foundation species in almost all communities (*Haloin and Strauss, 2008*). Community genetics is a basic level of investigation—meaning it is close to the basic unit acted on by evolution, the gene—within the larger effort to meet what *Haloin and Strauss (2008)* refer to as 'one of the supreme challenges of biology': understanding how the feedback loops between community ecology and evolution shape biodiversity.

**eLife digest** Plants are at the base of many food webs. This means that the different traits and characteristics of the plant species in an ecosystem can have a large impact on the animals and other organisms that live there.

Individuals within the same plant species often differ in multiple genes. This 'genetic diversity' can affect the populations of other organisms in their ecosystem, for example, by altering which species are present, and the number of individuals. However, previous studies that investigated plant genetic diversity in ecosystems used plants that varied in multiple, usually unknown, genetic traits, which made it difficult to identify specific genetic traits in plants that can influence the whole ecosystem.

One way that plants affect their ecosystems involves how they defend themselves against the herbivores that try to eat them. For example, plants can use sharp spines or harmful chemicals to deter herbivores, or attract predators that will attack the herbivores.

Here, Schuman et al. carried out a 2-year field study using wild tobacco plants that had been genetically altered to employ different defensive strategies. This was achieved by altering the expression of three genes in the plants in specific combinations. Two of the genes, called *LOX2* and *LOX3*, are required to make most of the chemicals that tobacco plants use to defend themselves against herbivores. The third gene, called *TPS10*, which comes from the crop plant maize, gives plants the ability to release a fragrance that attracts natural predators of their herbivores.

Except for these specific alterations, the plants were otherwise genetically identical. These plants were then grown either alone, or in groups of five plants, which reflects the normal size of groups of wild tobacco growing in its natural environment. The groups contained mixtures of plants with different gene alterations.

Schuman et al. found that the expression levels of these genes in individual plants could determine how well the whole group fared in several different measures of plant health and defense. For example, plants lacking *LOX2* and *LOX3* usually appeared to be less healthy and they were less likely to survive long enough to reproduce because they were less able to defend themselves against herbivores. However, if these plants were grown in a group with one plant that expressed *TPS10*, all of the plants in the group looked healthier and were more likely to survive long enough to reproduce.

Many crops are grown in large fields containing individual plants that are largely genetically identical, which makes them more vulnerable to disease. Schuman et al.'s findings suggest that some of the community-level protective effects provided by diversity in wild populations of plants could be introduced into crop fields by altering the expression of a few specific genes in some individuals.

The maintenance of biodiversity and the ecosystem services it supports is a matter of increasing concern as the human population grows, and agricultural resource use increases (*Green et al., 2005*). In a meta-analysis of 446 biodiversity measures taken between 1954 and 2004, of which 312 manipulated plant species biodiversity, species-level biodiversity was shown to positively affect several ecosystem properties including stability, regulation of biodiversity, nutrient cycling, erosion control, and productivity at the primary, secondary, and tertiary levels (*Balvanera et al., 2006*). Functional differences among species are often thought to be the source of effects resulting from species-level biodiversity (*Ebeling et al., 2014*). In fact, a growing number of studies has shown that genetic diversity within species can have large effects on ecological community composition (e.g., *Crutsinger et al., 2006*; *Johnson et al., 2006*; *Cook-Patton et al., 2011*). *Whitham et al. (2012)* refer to the genetically based tendency for particular genotypes to support different ecological communities as 'community specificity'. Some, or even most, biodiversity effects may be explained through the community specificity of particular genetic traits.

Four postulates of community genetics provide the roadmap for understanding the community effects of genes (*Wymore et al., 2011*; *Whitham et al., 2012*): (1) the focal species must have a significant effect on its community or ecosystem, (2) the trait under investigation must be genetically

based and heritable, (3) conspecifics differing genetically in this trait must have quantifiably different effects on community and ecosystem processes, and (4) a predictable effect on those community and ecosystem processes must follow when the gene of interest is manipulated. As Koch's postulates first allowed medical researchers to identify disease-causing agents, these postulates allow community geneticists to identify genes responsible for structuring specific aspects of ecological communities. So far, although postulates one through three have been satisfied in several ecological systems, postulate four—the critical step in attributing community functions to specific genes—has been tested in only a handful of systems (*Whitham et al., 2006*; *2012*). *Whitham et al. (2012)* suggest that improved genetic tools will soon permit the widespread testing of postulate four, as has been done for nicotine in the wild tobacco *Nicotiana attenuata*: an ecological model plant with a transformation system and a well-characterized ecological community (*Krügel et al., 2002*; *Steppuhn et al., 2004*; *Kessler et al., 2008*; reviewed in; *Schuman and Baldwin, 2012*).

In this study, we built on deep molecular ecological knowledge of *N. attenuata* to demonstrate that all four postulates for genes controlling a suite of ecologically characterized defense traits (*Kessler and Baldwin, 2001*; *Kessler et al., 2004*; *Allmann and Baldwin, 2010*; *Schuman et al., 2012*). Specifically, we manipulated direct defenses: traits which directly reduce plant fitness loss to herbivores, and indirect defenses: traits which attract enemies of herbivores which remove or impair herbivores, and thereby indirectly reduce plant fitness loss to herbivores. *N. attenuata*'s direct and indirect defenses, their biosynthesis and ecological functions have been characterized in detail over two decades of research (reviewed in *Schuman and Baldwin, 2012*). The direct defenses comprise a suite of chemical traits, most of which are induced by herbivory via jasmonate signaling, and genetically silencing these defenses causes significant changes in herbivore communities (*Kessler et al., 2004*; *Gaquerel et al., 2010*; *Kallenbach et al., 2012*). *N. attenuata* also produces a complex blend of herbivore-induced plant volatiles (HIPVs) which can be parsed into terpenoids, GLVs and other fatty acid derivatives, and aromatics, and are regulated by elicitors in herbivore oral secretions (OS) (*Gaquerel et al., 2009*; *Schuman et al., 2009*). Of these, the jasmonate-regulated sesquiterpene (*E*)-α-bergamotene (TAB) and the GLVs have been shown to reduce herbivore loads by attracting predators, which for the GLVs has also been shown to result in greater plant fitness (*Halitschke et al., 2000*; *Kessler and Baldwin, 2001*; *Halitschke et al., 2008*; *Schuman et al., 2009*; *2012*; *Allmann and Baldwin, 2010*). The majority of *N. attenuata*'s direct and indirect chemical defenses can be abolished by silencing the expression of two biosynthetic genes: LIPOXYGENASE 2 (*LOX2*), which provides substrate for GLV biosynthesis; and *LOX3*, essential for jasmonate biosynthesis, and thus also for the expression of most direct defense traits and the emission of TAB (*Kessler et al., 2004*; *Allmann et al., 2010*).

We specifically manipulated the expression of genes controlling direct and indirect defense traits in a factorial design by combining the silencing of *LOX2* and *LOX3* with the overexpression of *Zea mays TERPENE SYNTHASE 10* (*TPS10*), which produces TAB and (*E*)-β-farnesene (TBF) as its main products (*Köllner et al., 2009*; *Schuman et al., 2014*). This allowed us to independently manipulate total direct and indirect defenses, and the specific volatiles, TAB and TBF. We grew these transgenic, otherwise isogenic genotypes during two consecutive field seasons, both in replicated populations of different compositions, and as individuals.

Studies of genetic diversity and community specificity most commonly measure the diversity and composition of invertebrate communities on plants, but also invertebrate abundance, and occasionally functional measures such as predation rates (reviewed in *Haloin and Strauss, 2008*). We focused on correlates of plant fitness: plant size, apparent health, flowering, and mortality before reproductive maturity; and on functional measures of the effectiveness of plant defense: total canopy damage from herbivores, herbivore abundance, and predation services received by plants in populations. The functional measures were chosen to reflect our observation of plants' ecological communities in both years. Nearly, all of these measures of plant fitness and interactions with herbivores and their predators were significantly affected not only by the genotype of individual plants, but by neighboring genotypes in populations.

## Results

Data from the study was deposited in Dryad (10.5061/dryad.qj007, *Schuman et al., 2015*).

## Genetically engineered phenotypes are maintained in experimental populations

Transgenic lines manipulated in the emission of TAB, TBF, and GLVs, and the production of total jasmonate-mediated defenses (*Figure 1* and associated source data files) were screened and characterized as described in Appendix 1. To investigate the community-level effects of variation in TAB and TBF in well- and poorly defended plants, we used four genotypes: WT plants, which have jasmonate-mediated induced direct defense compounds and herbivore-induced volatile emission comprising primarily sesquiterpenes and GLVs; *TPS10* plants, which are like WT but have both enhanced induced, as well as constitutive emission of the herbivore-induced sesquiterpenes TAB and TBF (*Schuman et al., 2014*); *lox2/3* plants, which are deficient in both jasmonate-mediated direct defense compounds and herbivore-induced volatiles, and *lox2/3xTPS10* plants, which are like *lox2/3*, but with constitutive emission of TAB and TBF.

These genotypes were grown as individuals and in five-plant populations in a field plot in *N. attenuata*'s native habitat (*Figure 2A*, *Figure 2—figure supplements 1*, *2*). A population size of five plants is realistic for natural populations of *N. attenuata* (*Figure 2—figure supplement 3*), particularly before 1995, after which an invasive brome grass fueled a dramatic increase in fire sizes and consequently *N. attenuata* population sizes. This choice of population size also allowed us to clearly differentiate between edge and center positions (*Figure 2B*). Populations were either monocultures, or mixed cultures containing a single *TPS10*-expressing plant at the center and plants of the same defense- and HIPV level at the edges (WT around *TPS10*, and *lox2/3* around *lox2/3xTPS10*, *Figure 2B*).

There was no effect of population type (individual, monoculture, mixed culture) or plant position (individual, edge, center) on genetically engineered levels of GLVs, jasmonate-mediated defense (trypsin protease inhibitor [TPI] activity served as an indicator), TAB and TBF (*Figure 1C*, *Figure 1—source data 3* and details in Appendix 2), with one exception: trace amounts of TAB were detected in open headspace trappings around young *lox2/3* plants at the edges of *lox2/3 + lox2/3xTPS10* mixed cultures, but not around *lox2/3* individuals or *lox2/3* plants at the edges of monocultures (*Figure 3*).

## LOX2 and LOX3 expression reduce foliar herbivore abundance and damage, while TPS10 expression further reduces herbivore abundance

Prior to the field experiment, we tested whether *TPS10* expression affects defense against a specialist herbivore in a WT or *LOX2/3*-deficient background via a no-choice assay with larvae of the naturally co-occurring solanaceous specialist *Manduca sexta* in the glasshouse. *M. sexta* larvae grew five times as large on *lox2/3* or *lox2/3xTPS10* plants as they did on WT or *TPS10* plants (*Figure 4*; Wilcoxon rank sum tests, Holm-Bonferroni-corrected p-values <0.001). In contrast, larvae grew to similar sizes on plants that differed only in the *TPS10* overexpression construct (corrected p-values >0.2). The progression of *M. sexta* larval instars was also faster on *LOX2/3*-deficient plants: all larvae feeding on *lox2/3* or *lox2/3xTPS10* were in the third instar by day 8, while 11/17 larvae feeding on WT and 12/15 larvae feeding on *TPS10* were still in the second instar (corrected p<0.001 in Fisher's tests for *lox2/3* v. WT, *lox2/3* v. *TPS10*, *lox2/3xTPS10* v. WT, and *lox2/3xTPS10* v. *TPS10*, corrected p=0.4440 for WT v. *TPS10*, and corrected p=1 for *lox2/3* v. *lox2/3xTPS10*).

In the field, total canopy damage from foliar herbivores was assessed on June 9th in season one, and June 2nd in season two. Canopy damage in both seasons was due to both generalist and specialist herbivores from chewing and piercing-sucking feeding guilds. The generalists included noctuid larvae, *Trimerotropis* spp. grasshoppers, *Oecanthus* spp. tree crickets, and leaf miner spp.; the specialists included the sphingid larvae *M. sexta* and *Manduca quinquemaculata*, flea beetles *Epitrix hirtipennis* and *E. subcrintia*, and the mirid *Tupiocoris notatus*. Canopy damage from herbivores was more than twice as great in season two (9–24% total canopy) as in season one (2–9% total canopy).

In both seasons, *lox2/3* and *lox2/3xTPS10* plants accumulated twice as much canopy damage as WT and *TPS10* plants, regardless of plant position or population type (*Figure 5* and associated source data files): there were no significant differences during either season between plants or populations differing only in *TPS10* expression, or plants of the same genotype in mono- vs mixed cultures with single *TPS10*-expressing plants; details of statistical tests are given in Appendix 3.

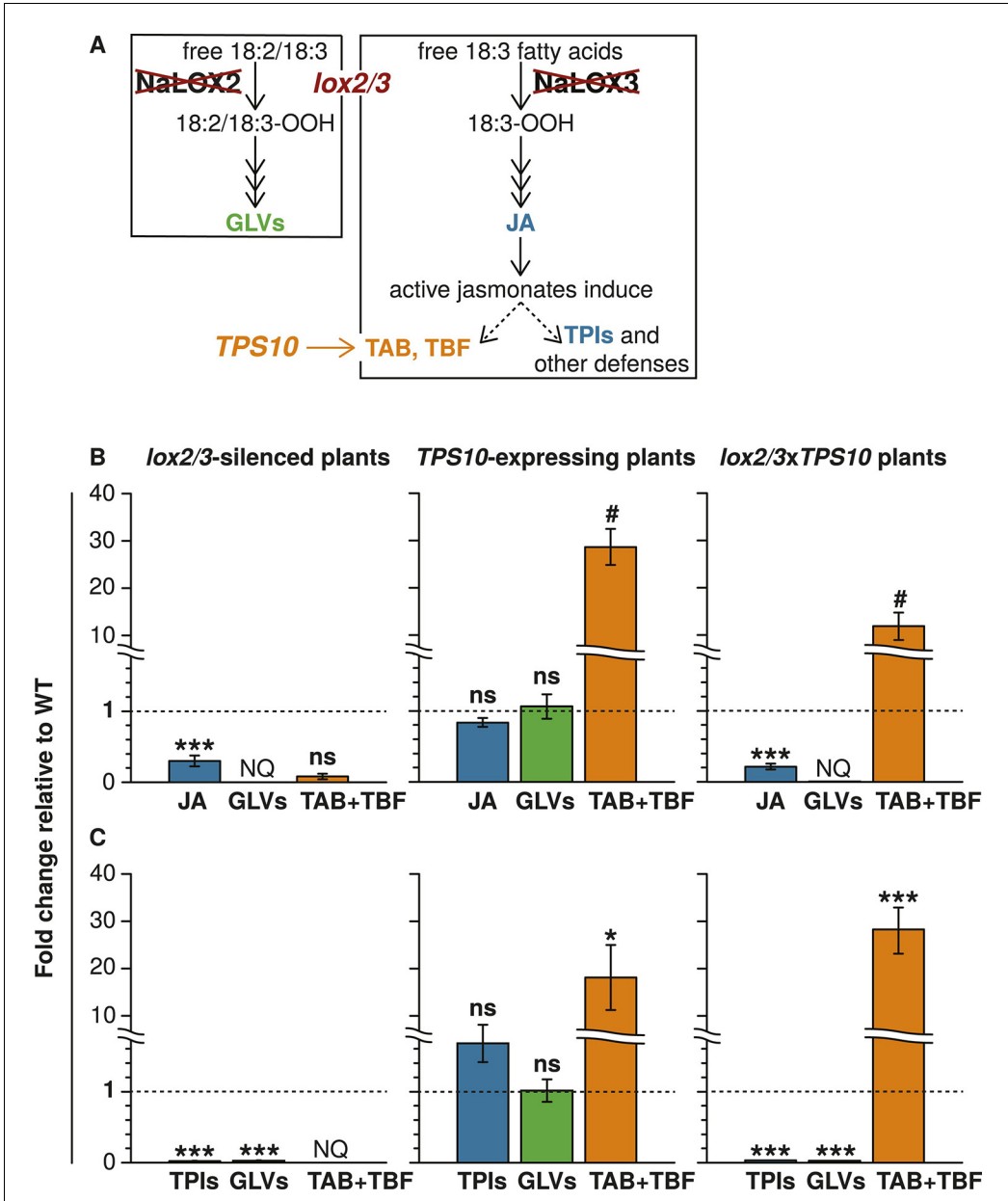

**Figure 1.** The *lox2/3* and *TPS10* transgenic constructs, alone and in combination (*lox2/3xTPS10*), independently alter herbivore-induced plant defenses and sesquiterpene HIPVs. (**A**) The endogenous lipoxygenase genes *LOX2* and *LOX3* and the *Z. mays* sesquiterpene synthase gene *TPS10* (*TPS10*) were manipulated in transgenic lines of *N. attenuata* in order to uncouple the production of two HIPVs—the sesquiterpenes (*E*)-α-bergamotene (TAB) and (*E*)-β-farnesene (TBF)—from jasmonate-mediated direct defenses and other volatiles. LOX2 and LOX3 provide fatty acid hydroperoxides from 18:2 and 18:3 fatty acids (18:2/18:3-OOH) for the synthesis of green leaf volatiles (GLVs) or jasmonic acid (JA) and the jasmonate-derived hormones, respectively. GLVs are released in response to herbivore damage, and jasmonates regulate the production of most anti-herbivore defenses, including trypsin protease inhibitors (TPIs), and other HIPVs. LOX2 and LOX3 were transgenically silenced using an inverted repeat (ir) construct specifically silencing both (*lox2/3*). Jasmonate-regulated volatiles in *N. attenuata* are mostly sesquiterpenes, of which the most prevalent is TAB, and include trace amounts of the biosynthetically related sesquiterpene TBF. The *Z. mays* sesquiterpene synthase *TPS10* (**Köllner et al., 2009**), which produces TAB and TBF as its main products, was ectopically over-expressed in *N. attenuata* plants under control of a 35S promoter (*TPS10*) to uncouple the emission of these volatiles from endogenous jasmonate signaling. (**B** and **C**) Plants with *lox2/3* and *TPS10* constructs, and a cross with both constructs having the transgenes in a hemizygous state (*lox2/3xTPS10*) had the expected phenotypes both in glasshouse and field experiments. For each compound or group of compounds, the mean WT value was set to 1 (indicated by dashed lines), and all values were divided by the WT mean resulting in mean fold changes ±SEM. A complete description of *TPS10* plants including the demonstration of a single transgene insertion and WT levels of non-target metabolites is provided in **Schuman et al. (2014)**, and additional information for *lox2/3*, *TPS10*, and *lox2/3xTPS10* plants (single transgene insertion for *lox2/3*, accumulation of target gene transcripts) is given in Appendix 1. (**B**) Glasshouse-grown plants were treated with wounding and *M. sexta* OS, and

*Figure 1 continued on next page*

*Figure 1 continued*

JA, GLVs, TAB and TBF were analyzed at the time of peak accumulation; n = 4. For *lox2/3* and *lox2/3xTPS10*, GLVs were not quantifiable (NQ) by GC–MS due to inconsistently detected or no detected signals. ***p<0.001 in Tukey HSD tests following a significant one-way ANOVA (p<0.001) of WT, *lox2/3*, and *lox2/3xTPS10*; # plants with the *TPS10* construct emit significantly more TAB + TBF (Holm-Bonferroni-corrected p<0.01 in a Wilcoxon rank sum test of WT and *lox2/3* vs *lox2/3xTPS10* and *TPS10*); ns, not significant. TBF was not detectable in *lox2/3* or WT samples. For absolute amounts of JA and other phytohormones, GLVs, TAB and TBF measured in glasshouse samples, see *Figure 1—source data 1*, *2*. (C) TPI activity, total detectable GLVs, and total TAB and TBF measured in frozen leaf samples harvested from field-grown plants at the end of experimental season two (June 28th), after plants had accumulated damage from naturally occurring herbivores. TPIs (n = 11–24) were slightly elevated in field samples of *TPS10* plants, but TPI activity did not differ from WT plants in glasshouse samples from two independent *TPS10* lines including the line used for this study (*Schuman et al., 2014*). Total GLVs, TAB, and TBF were quantified by GC–MS in hexane extracts from the same tissue samples used to measure TPI activity, n = 9–22 (a few samples had insufficient tissue for the analysis). Neither TAB nor TBF could be detected in *lox2/3* samples (NQ); TBF was also not detectable in WT samples. *Corrected p<0.05, ***corrected p<0.001 in pairwise Wilcoxon rank sum tests following significant (corrected p<0.05) Kruskal–Wallis tests across all genotypes for each category; p-values were corrected for multiple testing using the Holm-Bonferroni method; ns, not significant. For absolute amounts of TPIs, GLVs, TAB and TBF, and non-target volatiles measured in field-collected tissue samples, see *Figure 1—source data 3*; emission of GLVs, TAB, TBF, and non-target volatiles from field-grown plants is given in Appendix 2.

The following source data is available for figure 1:

**Source data 1.** Absolute values for JA and GLVs shown as fold-changes in *Figure 1B* in *lox2/3* and *lox2/3xTPS10* plants, as well as JA-Ile, individual GLVs, and leaf areas for headspace collection (mean ± SEM); n = 4.

**Source data 2.** Absolute values for TAB and TBF shown as fold-changes in *Figure 1B*, and leaf areas for headspace collection (mean ± SEM); n = 4.

**Source data 3.** Absolute values for analytes shown as fold-changes in *Figure 1C* (mean ± SEM).

Damage observed could be attributed to specific herbivores according to typical feeding patterns and the observation of feeding individuals. Individual herbivores generally followed the trend of total herbivore damage, causing greater damage on *lox2/3* and *lox2/3xTPS10* plants (*Figure 5—source data 1*, *2*). The largest and most significant differences (p-values < 0.05) were between plants with vs without the *lox2/3* silencing construct (*lox2/3* and *lox2/3xTPS10* v. WT and *TPS10*, statistical comparisons in *Figure 5—source data 1*, *2*). All significant differences were between plants or populations with vs without the *lox2/3* silencing construct, and never between plants or populations differing only in the *TPS10* overexpression construct, nor between plants of the same genotype in different population types (*Figure 5* and associated source data files). It should be noted that in season one, multiple lepidopteran species were present on plants and their damage could not be clearly distinguished; since these included generalist species as well as *Manduca* spp., lepidopteran damage in season one also could not be categorized as generalist or specialist damage (*Figure 5—source data 1*). In season two, lepidopteran damage not resulting from experimental infestation for predation assays (not counted toward total canopy damage) was due to noctuid species (*Figure 5—source data 2*).

The abundance of arthropods on plant shoots was quantified for a subset of plants on June 26th in season two, at which time *T. notatus* adults and nymphs and *Epitrix* spp. adults were the only foliar herbivores observed. Interestingly, total herbivore abundance at the end of the season did not entirely reflect foliar herbivore damage assessed on June 9th (*Figure 5B*), although a visual assessment of the plants indicated that relative herbivore damage levels still reflected the June 9th data: herbivores were less abundant on plants expressing *TPS10*, independently of *lox2/3* expression (*Figure 6A* and associated source data files). Herbivore abundance on WT plants differed depending on plants' location: in a comparison of abundance on individuals vs plants at the edges or centers of monocultures, abundance was highest on individuals and lowest on center plants. In contrast, herbivore abundance on *TPS10*-expressing plants did not depend on plant location (*Figure 6B*, Appendix 3). Foliar herbivore abundance on WT and *lox2/3* edge plants was not affected by the presence of a *TPS10*-expressing neighbor: neither population type (monoculture or mixed culture), nor the interaction of genotype with population type was significant. However for WT and *lox2/3* edge plants, herbivore abundance did vary as a function of genotype, with abundance indeed being greater on *lox2/3* edge plants (*Figure 6C*, Appendix 3), consistent with greater damage to these plants (*Figure 5D*).

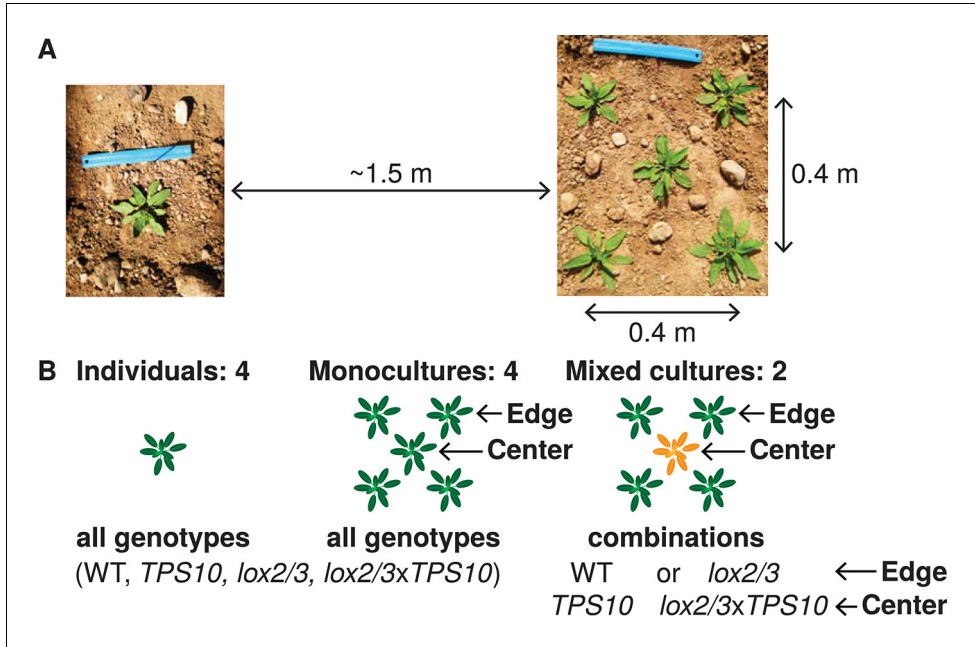

**Figure 2.** Field experiments designed to measure genotype-by-population effects of TPS10 volatiles in well-defended (WT, *TPS10*) or poorly defended (*lox2/3, lox2/3xTPS10*) plants. Two similar experiments were conducted in two consecutive field seasons. (**A**) Each genotype was planted as individuals and in five-plant populations (see *Figure 2—figure supplement 1*). Each individual or population was planted ca. 1.5 m from the nearest neighboring individual/population, measured perimeter-to-perimeter. Populations consisted of five plants arranged as shown in a 0.4 × 0.4 m square. (**B**) Individual and population types: monocultures and, additionally, mixed cultures were planted in which a *TPS10* plant (*TPS10* or *lox2/3xTPS10*) was surrounded by four plants of the same level of jasmonate-mediated defense and GLVs (WT or *lox2/3*). Replicates (n = 12 in season one, 15 in season two of each individual or population type) were arranged in a blocked design (see *Figure 2—figure supplement 2*): blocks consisting of one replicate of each type (all six randomly-ordered populations followed by all four randomly-ordered individuals) were staggered such that consecutive blocks were not aligned, and thus populations and individuals were also interspersed. Random order was modified only when necessary to ensure that no two replicate individuals/populations were placed next to each other vertically, horizontally, or diagonally. This planting design reflects typical distributions in native populations of *Nicotiana attenuata* (see *Figure 2—figure supplement 3*), particularly before 1995, after which an invasive brome grass fueled a dramatic increase in fire sizes and consequently *N. attenuata* population sizes.

The following figure supplements are available for figure 2:

**Figure supplement 1.** Photographs of the field experiment in season one.

**Figure supplement 2.** Layout in experimental season two, and color codes.

**Figure supplement 3.** Example of plant distribution in a native *N. attenuata* population, photographed in 2004.

---

Thus *LOX2* and *LOX3* expression increased plants' susceptibility to foliar herbivores, measured as herbivore growth or percentage of plant canopy damaged (*Figures 4*, *5*), but the relationship between foliar herbivore abundance and *LOX2/LOX3* expression was more complex and dependent on plants' position in populations (*Figure 6*). In contrast, herbivore abundance, but not damage or performance, was consistently reduced on *TPS10*-expressing plants (*Figures 4–6*).

### *LOX2/3* deficiency accelerates flowering under low herbivory, while *TPS10* expression reduces flower production under high herbivory

We monitored plant growth and size, to control for these non-defense traits which could confound the analysis of differences among genotypes, plant positions, and population types, and their

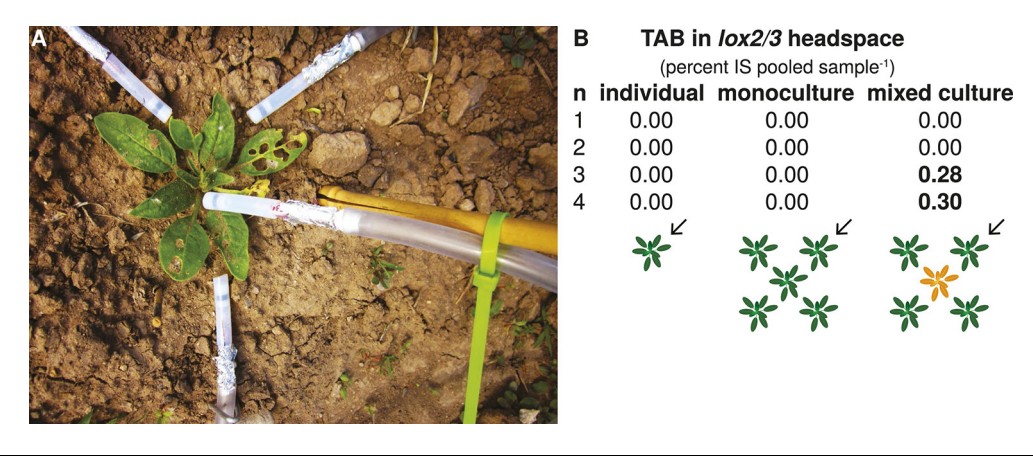

**Figure 3.** The TPS10 product TAB can be detected in the open headspace around *lox2/3* plants in mixed cultures containing a *lox2/3xTPS10* plant. (**A**) An 'open headspace' trapping was conducted during season two (May 16th) with rosette-stage individuals and edge plants as shown, using activated charcoal filters shielded from ozone and UV by $MnO_2$-coated copper ozone scrubbers (black strips in Teflon tubes pointed at plant) and aluminum foil, respectively. The headspace was sampled during the day, when sesquiterpenes are most abundant (see Appendix 2). Eluents from all four filters were combined and analyzed using highly sensitive GCxGC-ToF analysis; n = 4 plants. (**B**) The *TPS10* product *trans*-α-bergamotene (TAB) could be detected in the open headspace of two of four *lox2/3* plants at the edges of mixed cultures containing a *lox2/3xTPS10* plant, but not in *lox2/3* individuals or *lox2/3* plants in monocultures. Peak areas were normalized to the internal standard (IS) peak. Arrows indicate the plant that was sampled.

interactions with insect communities. Rosette diameter and plant stem height were measured at three times during the main period of vegetative growth in season one (May 1st–2nd, 8th–11th, and 16th–17th), and after plants had transitioned from vegetative growth to reproduction in season two

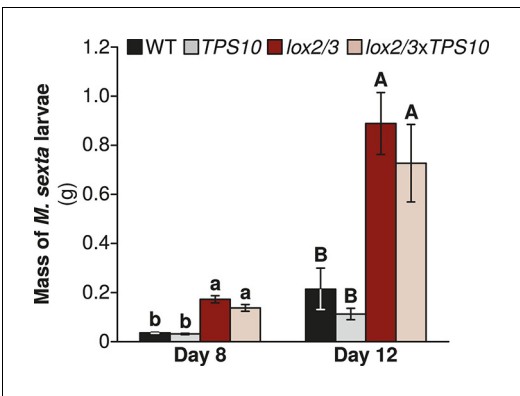

**Figure 4.** *Manduca sexta* larvae grow larger on *LOX2/3*-deficient plants, regardless of *TPS10* expression, in a glasshouse experiment. Data are shown as mean ± SEM; n = 15–19 larvae on day 8 and 11–16 larvae on day 12. One *M. sexta* neonate per plant (starting n = 25 larvae) was placed immediately after hatching on the youngest rosette leaf of an elongated plant. Larvae were weighed at the third and fourth instars (of 5 total), corresponding to days 8 and 12, after which larvae become mobile between plants. [a,b/A,B] Different letters indicate significant differences (corrected p<0.001) in larval mass on different plant genotypes within each day, in Wilcoxon rank sum tests following significant (corrected p<0.001) Kruskal–Wallis tests for each day (WT vs *lox2/3* day 8, $W_{17,19}$ = 310, corrected p<0.001, day 12, $W_{15,15}$ = 217, corrected p<0.001; *TPS10* vs *lox2/3xTPS10*, day 8, $W_{15,20}$ = 290, corrected p<0.001, day 12, $W_{11,14}$ = 143, corrected p<0.001; WT vs *TPS10* day 8, $W_{17,15}$ = 95, corrected p=0.227, day 12, $W_{15,11}$ = 56, corrected p=0.361; *lox2/3* vs *lox2/3xTPS10* day 8, $W_{19,20}$ = 249, corrected p=0.201, day 12, $W_{16,14}$ = 134, corrected p=0.377). p-values were corrected for multiple testing using the Holm-Bonferroni method.

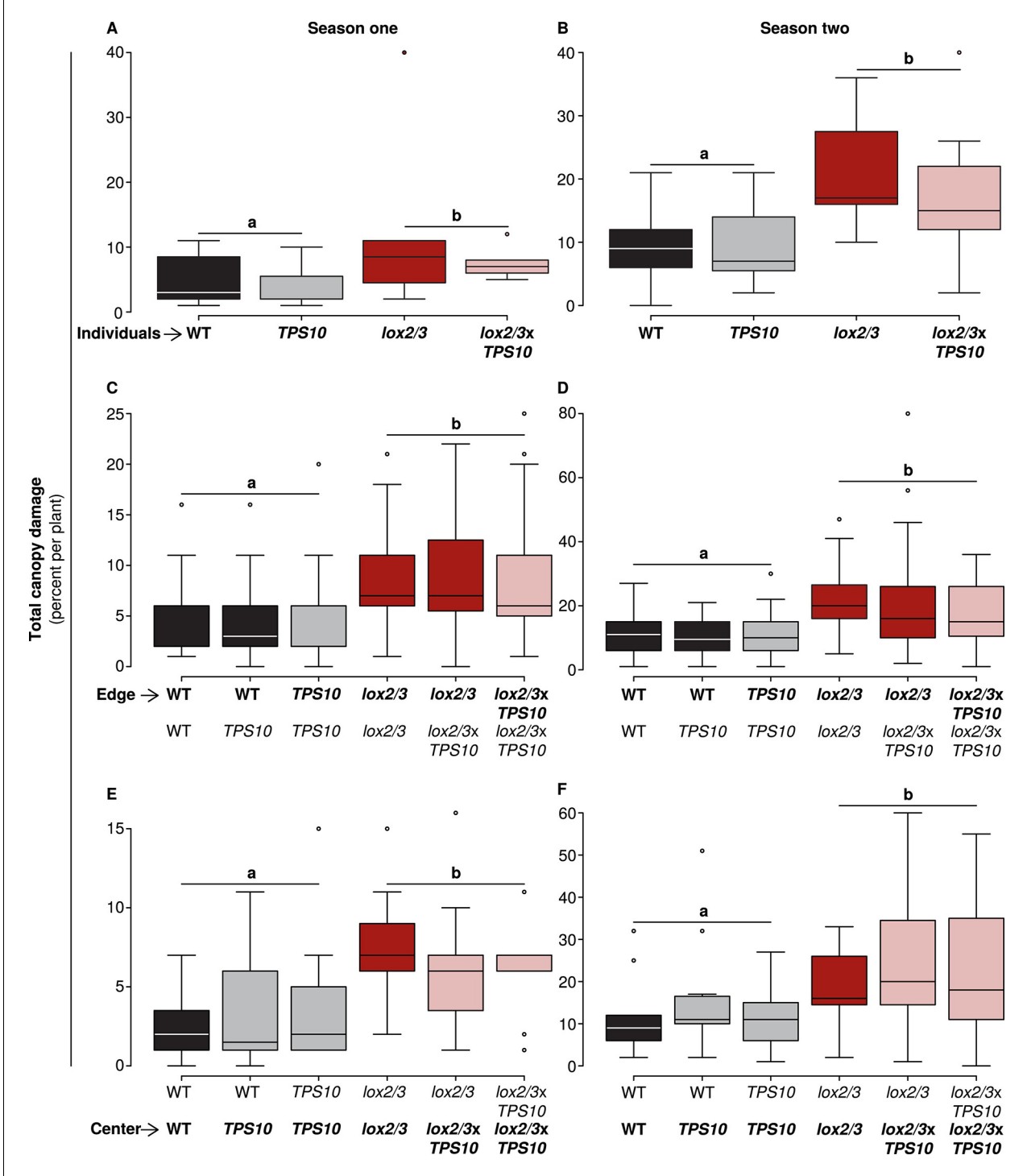

**Figure 5.** Only *LOX2/3* and not *TPS10*, plant position, or population type determined total foliar damage from herbivores over two consecutive field seasons. Note differences in y-axis scale. Data were collected on June 9th in season one (left panels), and on June 2nd in season two (right panels) from plants at all different positions in the experiment: individual plants (A–B), and plants at the edges (C–D) or at the centers (E–F) of populations. Black bars denote WT, grey bars *TPS10*, red bars *lox2/3*, and pink bars *lox2/3xTPS10* plants. Total canopy damage reflected the trends in canopy damage caused by individual herbivores (*Figure 5—source data 1*, *2*). Total canopy damage in season one was low (left panels), ranging from 5–10% on *lox2/3* and *lox2/3xTPS10* plants, and only 2–5% on WT and *TPS10* plants. Damage levels were about twice as high in season two (right panels), but showed the same relative pattern, with *LOX2/3*-deficient plans having more damage. Damage levels within a season were similar for plants in different positions (p>0.07, see Appendix 3). [a,b] Different letters indicate significant differences (p<0.05) between *LOX2/3*-expressing and *LOX2/3*-deficient plants (WT, *TPS10* v. *lox2/3*, *lox2/3xTPS10*) in minimal ANOVA or linear mixed-effects models on arcsin-transformed data. For individuals and center plants n = 7–

*Figure 5 continued on next page*

*Figure 5 continued*

13, and for edge plants n = 16–48 (up to 4 per population; the blocking effect was accounted for by a random factor in statistical analysis); exact replicate numbers are given in *Figure 5—source data 1*, *2*. There were no significant differences in damage between plants differing only in *TPS10* expression (p>0.4), plants in mono- vs mixed cultures (p>0.5), or plants of the same genotype at different positions (individual, edge, or center, p>0.07). Statistical models are given in Appendix 3.

The following source data is available for figure 5:

**Source data 1.** Total canopy damage from individual herbivores or groups of herbivores in experimental season one, corresponding to data shown in *Figure 5A* (mean ± SEM).

**Source data 2.** Total canopy damage from individual herbivores or groups of herbivores in experimental season two, corresponding to data shown in *Figure 5B* (mean ± SEM).

---

(June 4th–6th), at which time stem diameter was also measured. There were no significant differences in rosette diameter or stem diameter (p-values >0.08 in ANOVAs or linear mixed-effects models with LOX2/3 deficiency and TPS10 expression, plant position, and population type as factors: see statistical approach in Appendix 3). There were few differences in final measurements of plant height. *LOX2/3*-deficient plants were slightly taller at the final size measurement on May 17th of season one: the stem height of WT plants was 28.5 ± 1.1 cm, *TPS10* plants 27.2 ± 1.5 cm, *lox2/3* plants 35.4 ± 1.2 cm, and *lox2/3xTPS10* plants 32.2 ± 1.5 cm (mean ± SEM). In season two, final measurements of plant height on June 6th revealed only a minor difference for central plants expressing TPS10, discussed below. Statistical analyses of plant height are given in *Figure 8—source data 1*.

We also monitored early flower production by plants as a measure of growth and reproduction rates. To assess the rate of flowering, flowers were counted once shortly after plants first began to flower, and again 1 week later; in season two, plant size and flower production was additionally assessed once immediately prior to quantification of herbivore damage. In season one, in which herbivores damaged <10% of total canopy area (*Figure 5*), *LOX2/3*-deficient plants produced more flowers earlier than WT and *TPS10* plants. Shortly after the first plants began to flower in season one (May 11th), more *lox2/3* and *lox2/3xTPS10* than WT and *TPS10* plants were flowering (30% v. 14%), and this was still true 1 week later, after most plants had flowers (95% v. 79%; G-tests of independence, corrected p<0.001 on May 11th and 17th; *Figure 7A*). In pairwise comparisons among individual genotypes, there were no significant differences in G-tests of independence between WT and *TPS10* (corrected p-values > 0.2) or *lox2/3* and *lox2/3xTPS10* (corrected p-values >0.7). However, more *lox2/3* than WT plants were flowering at both timepoints (May 11th corrected p<0.001, May 17th corrected p=0.022), and more *lox2/3xTPS10* than *TPS10* plants were flowering on May 17th (corrected p<0.001). Both *lox2/3* and *lox2/3xTPS10* plants also produced more flowers per plant: on May 17th *lox2/3* had 6 (median = mean) flowers per plant while WT had 4, and *lox2/3xTPS10* had 4/5 (median/mean) flowers per plant while TPS10 had 2/3.

In season two, when herbivore damage ranged from 10–24% of total canopy (*Figure 5*), *LOX2/3*-deficient plants did not flower earlier (*Figure 7B*). Plants were in an earlier stage of flowering on May 29th in season two, than on May 17th in season one: both the proportion of flowering plants (72% May 29th season two, 87% May 17th season one) and the mean number of flowers per plant (4 on May 29th season two, 5 on May 17th season one) were lower for the May 29th observation. The differences between plants with or without the *lox2/3* silencing construct, which was evident on May 17th in season one, were not observed on May 29th in season two: the proportion of flowering plants was similar for all genotypes (68% for *lox2/3* and *lox2/3xTPS10* v. 75% for WT and *TPS10*, G-test of independence, corrected p = 1), as was the average number of flowers per plant: 1/3 (median/mean) for *lox2/3*, 3/5 for WT, 2/4 for *lox2/3xTPS10*, and 2/3 for *TPS10*. 1 week later, 100% of plants were flowering, and the median number of flowers per plant remained similar regardless of *LOX2/3* deficiency (*Figure 7B*).

However, in season two, there were subtler effects on flower production which could be attributed to plant position, *LOX2/3* deficiency, and *TPS10* expression (*Figure 8*). The plant height and flowering data collected immediately prior to the assessment of herbivore damage in season two were analyzed as described in Appendix 3 for effects of LOX2/3 deficiency, TPS10 expression, plant position, and population type (statistical analysis in the source data file for *Figure 8*). There were

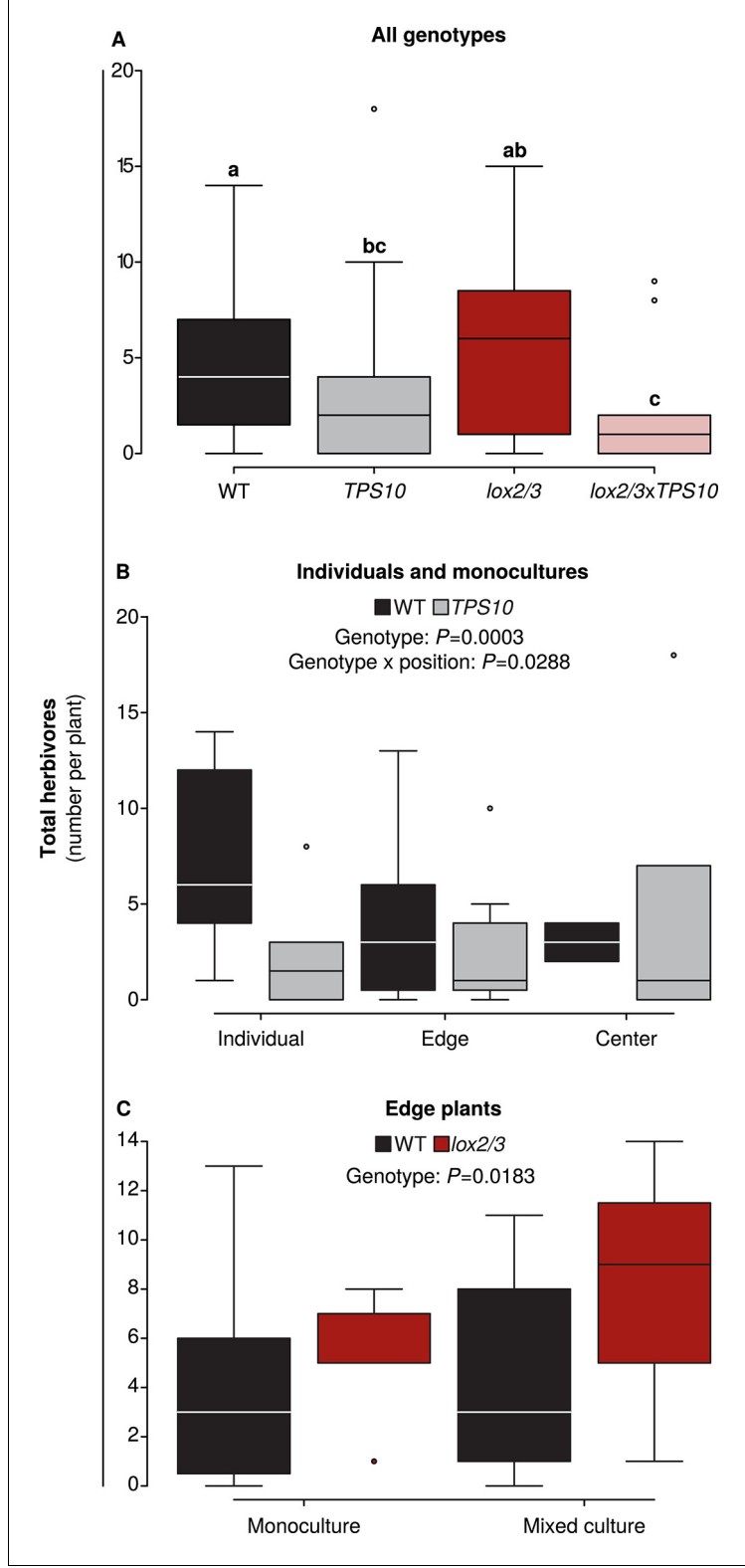

**Figure 6.** Foliar herbivore abundance is determined by *TPS10* expression, *lox2/3* expression, and plant position. Data were collected on June 26th in season two, when herbivores and predators were more abundant (see *Figure 5*). Black denotes WT, grey *TPS10*, red *lox2/3*, and pink *lox2/3xTPS10* plants. Counts for individual herbivores and predators are given in *Figure 6—source data 1*, *2*. (A) Herbivore counts on focal plants of each genotype revealed fewer herbivores present on plants expressing *TPS10* (n given in *Figure 6—source data 1*). [a,b]

*Figure 6 continued on next page*

*Figure 6 continued*

Different letters indicate significant differences (corrected p<0.05) between genotypes in Tukey contrasts following significant differences in a generalized linear mixed-effects model (see Appendix 3). (**B**) For WT and *TPS10* individuals or plants in monocultures (n given in **Figure 6—source data 2**), there was a significant effect (p<0.05) of plant genotype (z = −3.662, p=0.0003) and an interaction of genotype with position (*TPS10* by center vs individual position, z = 2.186, p=0.0288, generalized linear mixed-effects model in Appendix 3). (**C**) The presence of a *TPS10*-expressing neighbor in mixed populations did not significantly affect herbivore abundance on WT or *lox2/3* edge plants, but there was a significant difference between WT and *lox2/3* genotypes at the edges of populations (n given in **Figure 6—source data 2**, z = 2.358, p=0.0183, generalized linear model in Appendix 3).
The following source data is available for figure 6:

**Source data 2.** Total and median numbers of herbivores counted on focal plants of different genotypes on June 26th in season two, broken down into different population types and plant locations for WT and *TPS10* plants.
**Source data 1.** Total and median numbers of herbivores and predators counted on focal plants of different genotypes on June 26th in season two.

almost no differences in plant height (*Figure 8*, left panels), indicating that plant size differed little; the only significant difference was a small negative effect of *TPS10* expression for plants in the centers of populations, most pronounced in mixed cultures (*Figure 8E*). In contrast, flower production of individual plants was strongly decreased by *TPS10* expression and only slightly decreased by *LOX2/3* deficiency (*Figure 8B*), but differed little for plants in populations (*Figure 8*, right panels), with the exception that the shorter central plants also produced slightly fewer flowers (*Figure 8F*). Interestingly, for individual plants, there was an interactive effect of *TPS10* expression and *LOX2/3* deficiency resulting in similar flower production for *lox2/3xTPS10* plants and WT plants, although both *lox2/3* individuals and *TPS10* individuals produced significantly fewer flowers than WT (*Figure 8B*, statistical analysis in the source data file for *Figure 8*).

Overall, these data indicate that accelerated reproduction of *LOX2/3*-deficient plants in season one was due to a reduced net cost for these plants of *LOX2/3*-mediated defenses under low herbivore damage, which disappeared under higher herbivore damage in season two (*Figures 7*, *8* and associated source data file). In contrast, *TPS10* expression had little effect on plant growth and flowering in season one (*Figure 7*, source data file for *Figure 8*) and the effects of TPS10 expression on flowering in season two depended strongly on plant position (individual, edge or center, *Figure 8* and associated source data file). This indicates that *TPS10* expression did not have large, direct costs for plants in the field, but that the reproductive output of *TPS10* plants was sensitive to the presence and identity of neighbor plants. Especially under high herbivory in season two, there were few differences in plant size and these did not correspond to differences in herbivore damage (*Figures 5*, *8*).

## *LOX2/3* deficiency increases plant mortality, but mortality is mitigated by single *TPS10*-expressing plants in *LOX2/3*-deficient populations

We quantified plant mortality as an unambiguous measure of plant health and fitness (*Figure 9*). In our experiments, mortality can be interpreted as a measure of total reproductive potential and thus Darwinian fitness, because we observed mortality of plants before, or during, the peak of their flower production, and ended our experiments when plants would normally produce mature seed. (Production of viable seed is not generally allowed for field-released transgenic plants, and we did not allow our plants to produce seed.)

In season one, mean plant mortality was only 1%, or a total of 6 plants, by the end of experiments in mid-June. However, in season two, greater damage from herbivores (*Figure 5*) was associated with much higher mean plant mortality: 46%, or 236 plants by the end of experiments on June 29th. Overall, *LOX2/3*-deficient plants had higher mortality: 55% for *lox2/3* and *lox2/3xTPS10* vs 38% for WT and *TPS10* plants (G-test of independence, corrected p = 0.002). Thus, *lox2/3* plants in monocultures had significantly higher mortality than WT plants in monocultures (corrected p=0.003), and *lox2/3xTPS10* plants in monocultures had significantly higher mortality than *TPS10* plants in monocultures (corrected p=0.022) (*Figure 9*). In contrast, mortality rates for plants in mixed cultures were similar regardless of *LOX2/3* deficiency (corrected p=0.870). This was associated with marginally

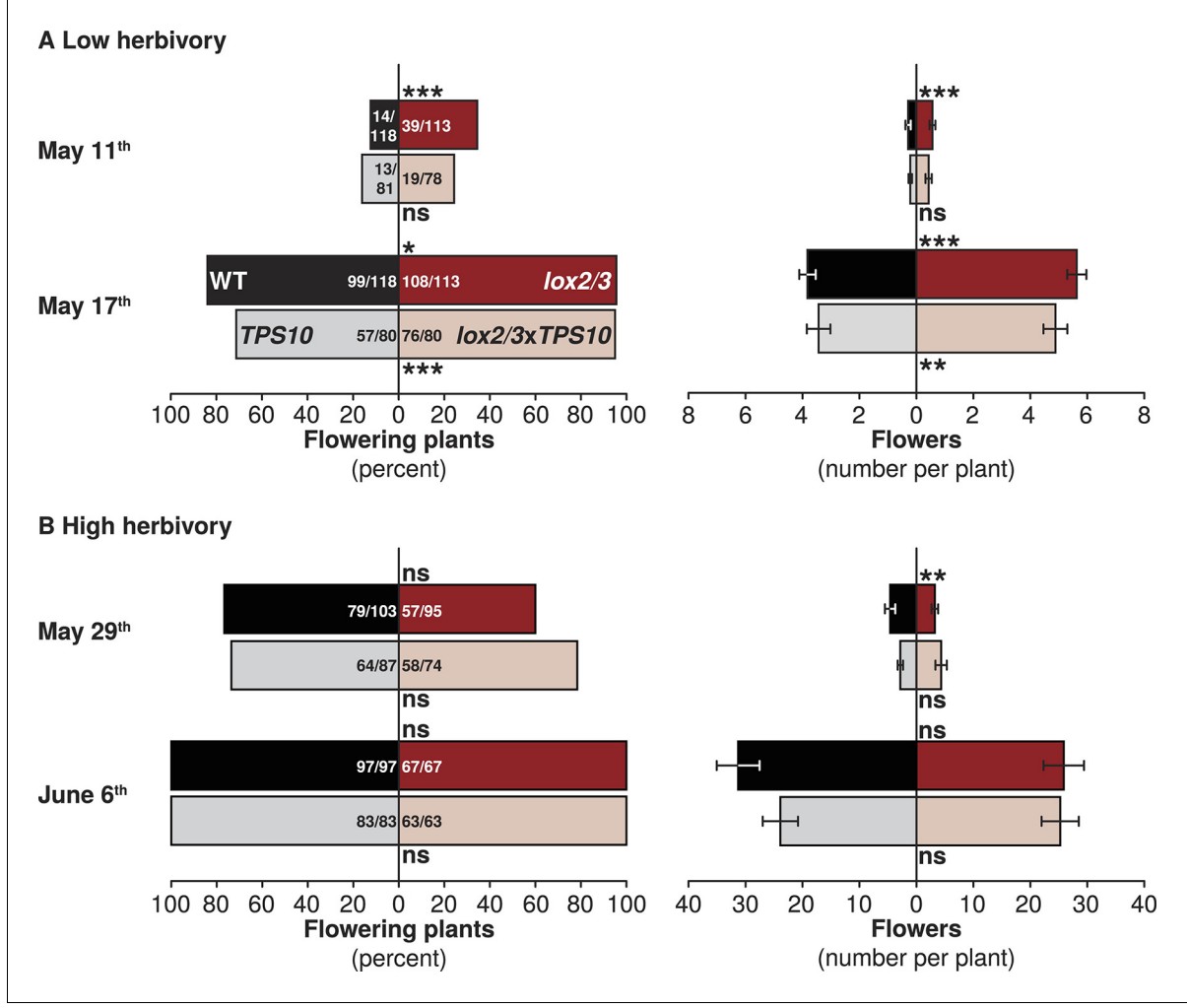

**Figure 7.** *LOX2/3* deficiency accelerated flowering under low herbivory. Flower production was monitored twice per season: once shortly after plants began to flower and again 1 week later, when most or all plants were flowering. Bars indicate either total percentage of plants of each genotype which were flowering (left panels), or the mean number of flowers per plant (right panels). WT and *TPS10* (black and grey bars) are shown opposite *lox2/3* and *lox2/3xTPS10* (red and pink bars). Ratios in bars indicate numbers of flowering/total plants monitored; a few plants were not included in counts which had recently lost their main stem to herbivores and had not yet re-grown. (**A**) In season one, when plants suffered less damage from herbivores (**Figure 5**), plants with the *lox2/3* silencing construct produced more flowers earlier: asterisks indicate pairwise differences between genotypes differing only in the *lox2/3* silencing construct. \*\*\*Corrected p<0.001, \*\*corrected p<0.01, \*corrected p<0.05, or no significant difference (ns) in G-tests (percentage flowering) or in Wilcoxon rank sum tests (flower number, WT vs *lox2/3*: May 11th, $W_{120,120}$ = 8582, p=0.0002; May 17th, $W_{120,119}$ = 5043, p<0.0001; *TPS10* vs *lox2/3xTPS10*: May 11th, $W_{84,84}$ = 3801, p=0.2063; May 17th, $W_{84,84}$ = 2520, p=0.0027). (**B**) In season two, during which plants received more damage from herbivores (**Figure 5**), flowering was monitored at later dates: the first timepoint in (**B**) is comparable to the second timepoint in (**A**) in terms of the proportion of plants flowering and the average number of flowers per plant (note difference in scale). In season two, *LOX2/3* deficiency rather decreased early flower numbers in the comparison of WT vs *lox2/3* on May 29th, and did not increase flower numbers at either measurement (WT vs *lox2/3*: May 29th, $W_{103,95}$ = 3860.5, p=0.0091; June 6th, $W_{97,69}$ = 2940.5, p=0.3018; *TPS10* vs *lox2/3xTPS10*: May 29th, $W_{87,74}$ = 3646.5, p=0.1417; June 6th, $W_{83,63}$ = 2695, p=0.7518). For a complete analysis of the effects of *LOX2/3* deficiency and *TPS10* expression, plant position, and population type on flowering and plant size, see source data file for **Figure 8**.

decreased mortality for plants in *lox2/3 + lox2/3xTPS10* mixed cultures (**Figure 9**, *lox2/3* mono v. *lox2/3 + lox2/3xTPS10* mix, corrected p=0.092). Interestingly, mortality in WT + TPS10 mixed cultures was slightly higher than in monocultures (45% v. 37% for WT monocultures and 31% for TPS monocultures), although these differences were also not significant (corrected p-values >0.090).

In summary, because plants in *lox2/3xTPS10* monocultures had higher mortality than those in *TPS10* monocultures, we conclude that expression of *TPS10* by *LOX2/3*-deficient plants did not compensate for the absence of *LOX2/3*-mediated defenses; yet surprisingly, single *lox2/3xTPS10* plants

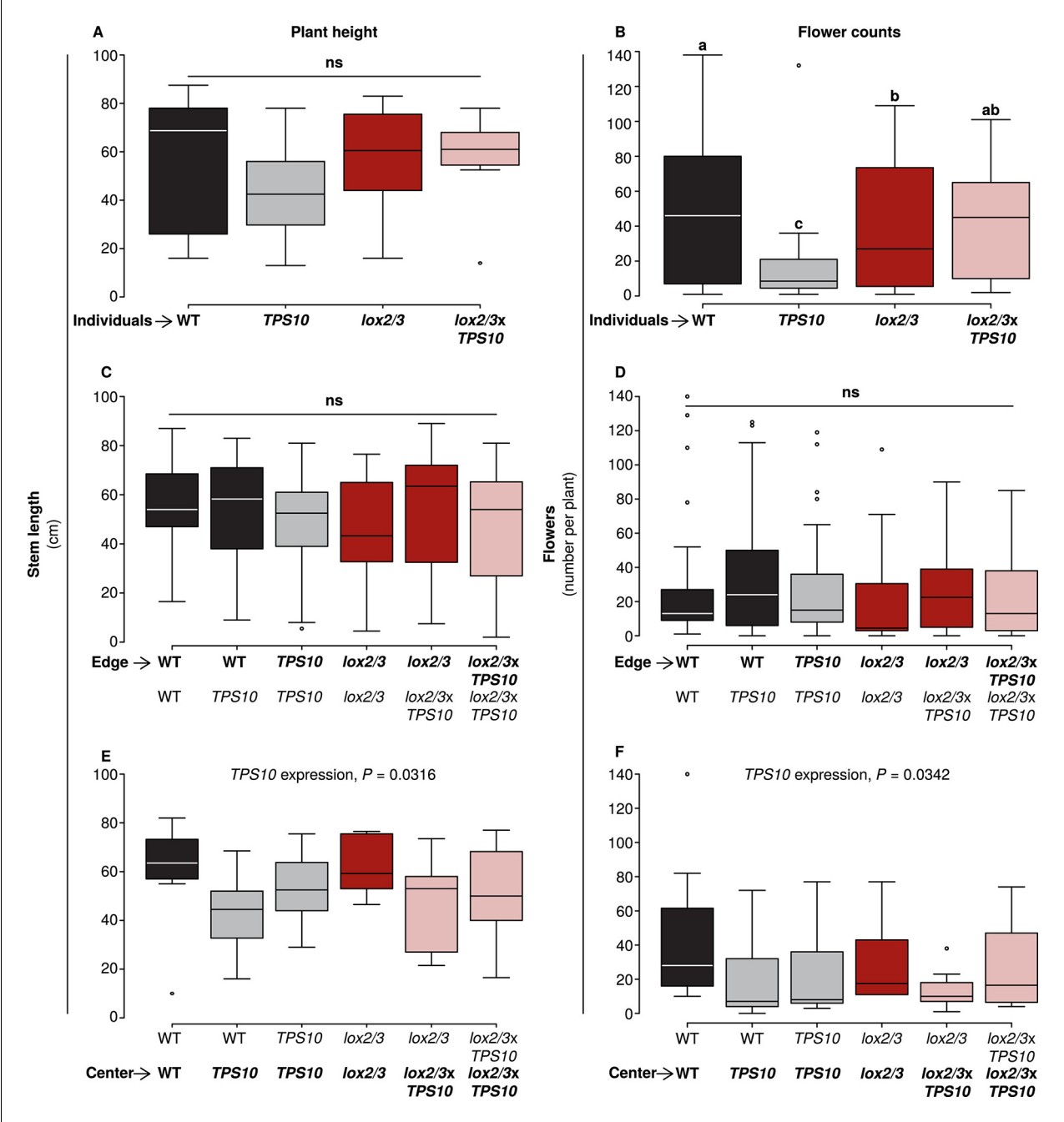

**Figure 8.** *TPS10* expression reduced flower production under high herbivory. Plant size as measured by stem length (left panels), and reproduction as measured by flower production (right panels); n given in *Figure 8—source data 1*. Data were collected on June 6th in season two from plants at all different positions in the experiment: individual plants (**A–B**), and plants at the edges (**C–D**) or at the centers (**E–F**) of populations. Black bars denote WT, grey bars *TPS10*, red bars *lox2/3*, and pink bars *lox2/3xTPS10* plants. Plant height differed little, with only a small negative effect of *TPS10* expression for center plants (**E**), which was more pronounced in mixed- than monocultures, indicating a slight competitive disadvantage for these plants. Individual plants differed much more than plants in populations in terms of flower production, with *TPS10*-expressing plants at a disadvantage (**B**); this effect was greatly reduced in populations.[a,b] Different letters indicate significant differences (corrected p<0.05) in Tukey post-hoc tests on plant genotype following significant effects in a generalized linear model, or generalized linear mixed-effects model (**C–D**) with genotype as a factor; these p-values were corrected using the Holm-Bonferroni correction for multiple testing as the same data were also tested for effects of *LOX2/3* deficiency and *TPS10* expression (see *Figure 8—source data 1*). There was no difference in flower production for edge plants, but a small negative effect of *TPS10* expression on flower production in center plants (**F**) corresponding to the slight reduction in height of these plants (**E**). Statistical models are given in *Figure 8—source data 1*.

*Figure 8 continued on next page*

*Figure 8 continued*

The following source data is available for figure 8:

**Source data 1.** The same statistical approach described in Appendix 3 for *Figure 5* was used for plant size and reproduction data from May 17th in season 1, and June 6th in season two (the most complete measurements, closest to the date of herbivore damage screens, from each season).

in *lox2/3* + *lox2/3xTPS10* mixed populations reduced the mortality of the entire population to the same level as in WT + *TPS10* mixed populations.

## *LOX2/3*-deficient plants have reduced apparent health, unless they have *TPS10*-expressing neighbors

Plant health, in terms of turgor, expanded leaf area and senescence, varied noticeably under the higher levels of herbivore damage in season two (*Figure 5*), and so plants were given an apparent health rating on a scale of one (dead) or two (low) to five (high). This scale was sufficiently objective that several observers were able to independently assign the same health index number to randomly-chosen plants, after training on fewer than 10 other plants. Ratings >1 on this health scale

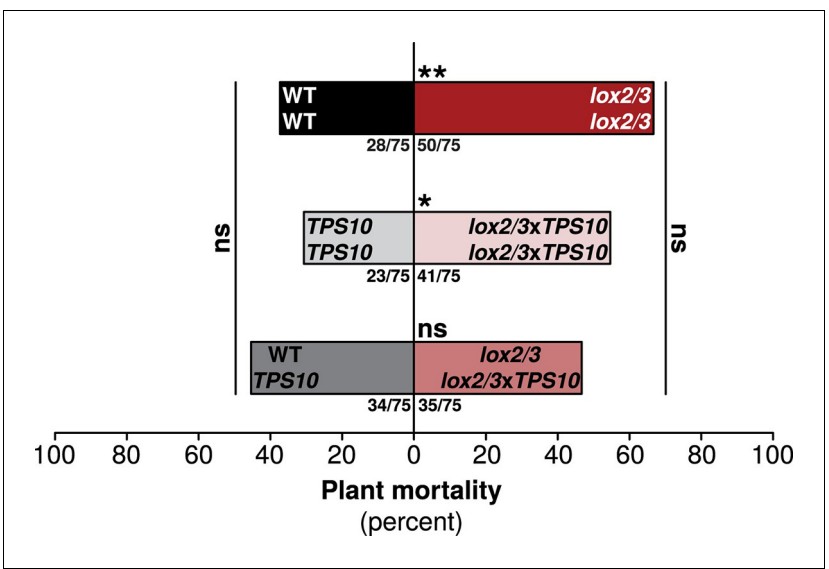

**Figure 9.** One *TPS10*-expressing plant per population counteracts a mortality increase in *LOX2/3*-deficient populations. Bars indicate total percentage mortality for all plants in each population type, and ratios in bars indicate numbers of dead/total plants (n = 75). WT and *TPS10* populations (left, black, and grey bars) are compared to the equivalent *lox2/3* and *lox2/3xTPS10* populations (right, red, and pink bars) differing only in the *lox2/3* silencing construct. *LOX2/3*-deficient plants had higher mortality in monocultures, regardless of whether they also expressed *TPS10* (**WT mono v. *lox2/3* mono, corrected p<0.01; *TPS10* mono v. *lox2/3xTPS10* mono, corrected p<0.05 in G-tests; p-values were corrected for multiple testing using the Holm-Bonferroni method). However, the mortality of plants in mixed *lox2/3* + *lox2/3xTPS10* populations is similar to that of the plants in mixed WT + *TPS10* populations (ns, not significant). There was no significant difference in mortality among mono- and mixed cultures of WT and *TPS10* plants (corrected p>0.3), or of *lox2/3* and *lox2/3xTPS10* plants, although there was a marginal difference between *lox2/3* monocultures and *lox2/3* + *lox2/3xTPS10* mixed cultures (corrected p=0.091). In addition, a health index was assigned to each plant (discussed in text); this index was correlated with several plant growth and reproduction parameters (see *Figure 9—figure supplement 1*). Data were collected on June 29th at the end of experimental season two; in season one, all plants experienced much lower herbivory (*Figure 5*) and negligible mortality (0–5%).

The following figure supplement is available for figure 9:

**Figure supplement 1.** The health index assigned to plants is strongly and significantly correlated to several measures of plant size and reproduction.

(living plants) were strongly correlated to several growth and reproduction traits measured (Spearman's ρ values ≥0.55, corrected p-values <0.001): number of buds, stem diameter, number of side branches, rosette diameter, stem length, and number of flowers (*Figure 9—figure supplement 1*).

*LOX2/3* deficiency was associated with significantly poorer apparent health on this scale: excluding dead plants, a median health index of 3 for *lox2/3* and *lox2/3xTPS10* vs 3.5 for WT and *TPS10* (Wilcoxon rank sum test, $W_{153,186}$ = 19,057.5, corrected p<0.001). In pairwise tests between genotypes for surviving plants, WT appeared significantly healthier than *lox2/3*, but not *lox2/3xTPS10* (WT v. *lox2/3*, $W_{102,82}$ = 2722, corrected p<0.001; WT v. *lox2/3xTPS10*, $W_{102,71}$ = 2687, corrected p=0.059); whereas *TPS10* appeared significantly healthier than both *lox2/3* and *lox2/3xTPS10* (*TPS10* v. *lox2/3*, $W_{84,82}$ = 1996.5, corrected p<0.001; *TPS10* v. *lox2/3xTPS10*, $W_{84,71}$ = 1995, corrected p=0.006). There was no difference in the apparent health of WT vs *TPS10* or *lox2/3* vs *lox2/3xTPS10* (corrected p-values=1, *TPS10* v. WT, $W_{84,102}$ = 4619; *lox2/3* v. *lox2/3xTPS10*, $W_{82,71}$ = 2604.5).

Separate examination of surviving individual, center, and edge plants revealed that *LOX2/3* deficiency was only associated with poorer apparent health in edge plants (*lox2/3* and *lox2/3xTPS10* v. WT and *TPS10*: individuals, $W_{21,22}$ = 230.5, corrected p=1; edge plants, $W_{102,129}$ = 9544, corrected p<0.001; center plants, $W_{30,35}$ = 598.5, corrected p=1). There were significant differences in the apparent health of edge plants in different population types: WT edge plants appeared significantly healthier than *lox2/3* plants at the edges of monocultures, but not healthier than *lox2/3* plants at the edges of *lox2/3* + *lox2/3xTPS10* mixed cultures (WT mono v. *lox2/3* mono, $W_{42,29}$ = 929, corrected p = 0.003; WT + *TPS10* mix v. *lox2/3* mono, $W_{40,29}$ = 891.5, corrected p=0.003; WT mono v. *lox2/3* +*lox2/3xTPS10* mix, $W_{42,34}$ = 953, corrected p=0.164; WT + *TPS10* mix v. *lox2/3* + *lox2/3xTPS10* mix, $W_{40,34}$ = 892, corrected p=0.249). However, *TPS10* plants at the edges of monocultures appeared significantly healthier than *lox2/3* or *lox2/3xTPS10* edge plants in mixed- or monocultures (*TPS10* mono v. *lox2/3* mono, $W_{47,29}$ = 1129.5, corrected p<0.001; *TPS10* mono v. *lox2/3* + *lox2/3xTPS10* mix, $W_{47,34}$ = 1209.5, corrected p=0.001; *TPS10* mono v. *lox2/3xTPS10* mono, $W_{47,39}$ = 1389.5, corrected p<0.001). No other comparisons between plants of individual population types

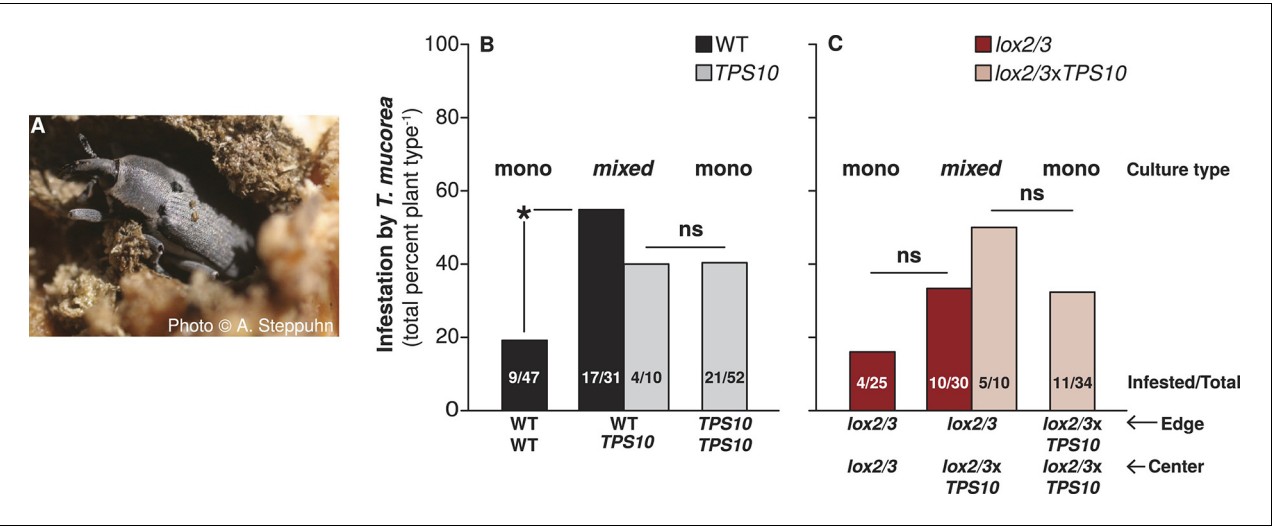

**Figure 10.** *TPS10* plants increase the infestation rates of their WT neighbors with the stem-boring weevil *T. mucorea*. (**A**) Photograph of a *T. mucorea* adult emerging from the stem which it infested as a larva reprinted with permission from Anke Steppuhn, Copyright 2005. All rights reserved. Adults mate and oviposit on young *N. attenuata* plants in March and April, and upon hatching, larvae burrow into the growing stem of the plant chosen by their mother (*Diezel et al., 2011*). Infestation was scored as the number of plants with hollow, frass-filled stems (from which larvae were usually also recovered). (**B**) *T. mucorea* infestation more than doubled for WT plants, but did not change for *TPS10* plants in mixed cultures vs monocultures. Bars indicate total percentage of plants infested; ratios in bars indicate numbers of infested/total plants. Data were collected at the end of experimental season two (June 29th). *Corrected p<0.05 in a Fisher's exact test, n = 31–47 (all surviving replicates). p-values were corrected for multiple testing using the Holm-Bonferroni method. (**C**) The presence of *lox2/3xTPS10* plants also tended to increase *T. mucorea* infestation in *lox2/3* + *lox2/3xTPS10* populations but, likely due to lower replicate numbers caused by higher mortality for plants in *lox2/3* and *lox2/3xTPS10* monocultures (*Figure 9*), differences are not significant (corrected p-values=1).

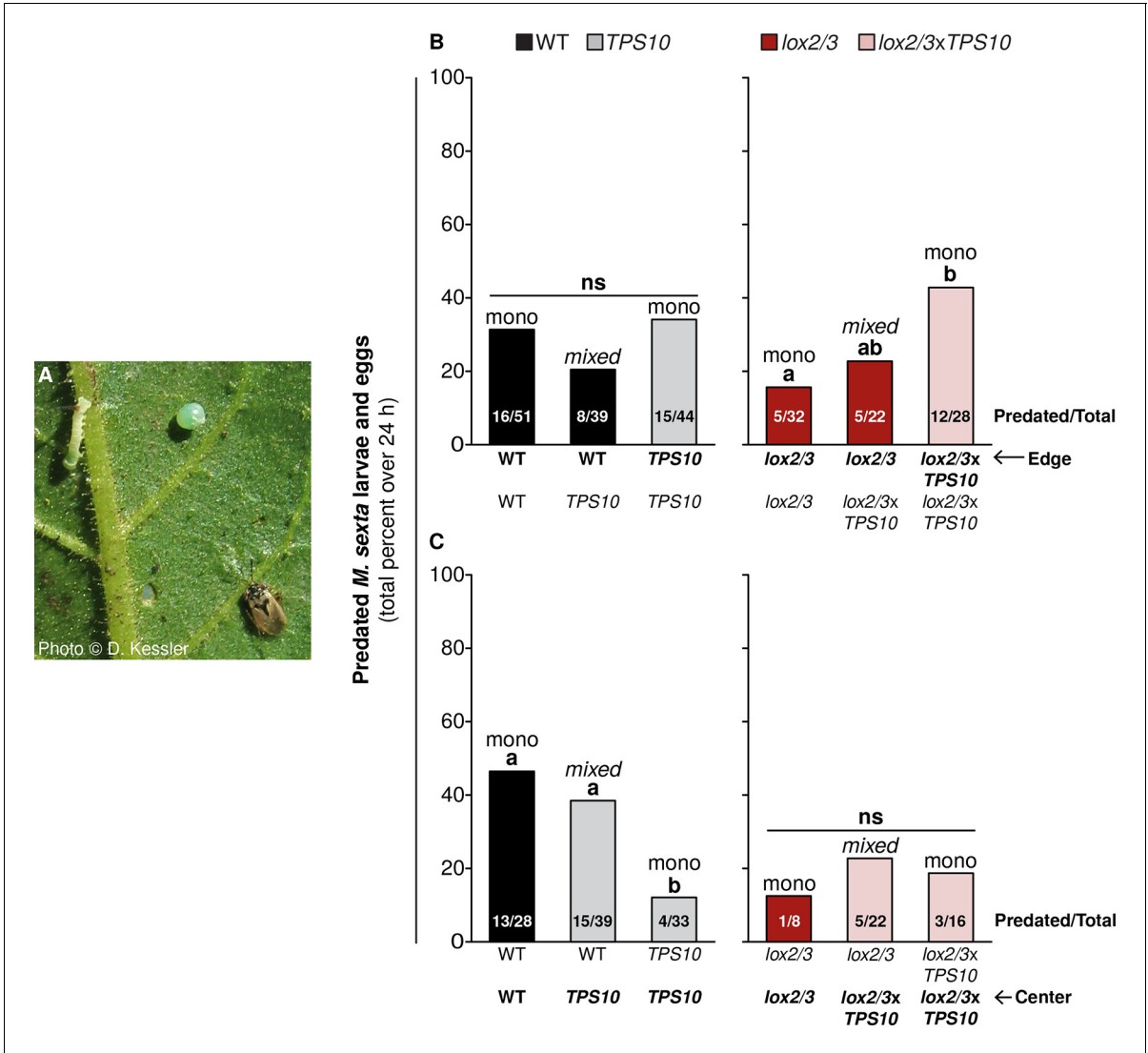

**Figure 11.** *TPS10* and *LOX2/3* interact to alter the predation of *M. sexta* by *Geocoris* spp from edge vs center plants. (**A**) Photograph of a *M. sexta* egg and first-instar larva, and *G. pallens* adult on an *N. attenuata* leaf reprinted with permission from Danny Kessler, Copyright 2006. All rights reserved. As shown here, one first-instar *M. sexta* larva and one egg were placed on a lower stem leaf in a standardized position on healthy, size-matched plants (one leaf/plant), and their predation was monitored for 24 hr. Data are total numbers from five consecutive trials conducted between June 21st and June 27th at the end of experimental season two, when *Geocoris* spp. and herbivores were more abundant (see *Figure 5*). (**B** and **C**) Predation rates of *M. sexta* are higher on *lox2/3xTPS10* than *lox2/3* plants at the edges of monocultures (**B**), but lower on *TPS10* than WT plants at the centers of monocultures (**C**). Bars indicate the total percentage of *M. sexta* eggs and larvae predated per population; numbers of predated/total eggs and larvae are given inside bars. The same trends were observed in both egg and larva predation; separate numbers of larvae and eggs predated are given in *Figure 11—source data 1*. Plants of the WT background and plants with the *lox2/3* silencing construct were separately matched for parallel experiments and thus analyzed separately. (**B**) Predation of herbivores from edge plants was similar (not significant, ns) regardless of the number of *TPS10* plants in WT + *TPS10* populations, but increased with increasing numbers of *lox2/3xTPS10* plants in *lox2/3* + *lox2/3xTPS10* populations. For plants at the center of populations, the pattern was reversed: (**C**) predation rates decreased with increasing numbers of *TPS10* plants in WT + *TPS10* populations, and were similar (ns) regardless of the number of *lox2/3xTPS10* plants in *lox2/3* + *lox2/3xTPS10* populations. [a,b] Different letters indicate significant differences (corrected $p < 0.05$) in Fisher's exact tests; p-values were corrected for multiple testing using the Holm-Bonferroni method.

The following source data is available for figure 11:

**Source data 1.** Total numbers of *M. sexta* larvae and eggs predated by *Geocoris* spp. in experimental season two (corresponding to data shown in *Figure 10*).

were significant (corrected p-values >0.08) and there was no significant association between apparent health and plant position (Kruskal–Wallis test across individual, edge, and center plants, $\chi^2_2 = 2.38$, corrected p=1) or plants of the same genotype in mono- vs mixed cultures (Wilcoxon rank sum tests, corrected p-values=1).

In summary, as for foliar herbivore damage, growth, flowering, and mortality, there was no significant difference in the apparent health of plants differing only in *TPS10* expression. However, *TPS10* expression in *lox2/3xTPS10* plants was associated with an apparent health similar to WT plants, while *lox2/3* plants had a significantly lower apparent health than WT plants. Furthermore, the differences in apparent health were stronger between *TPS10* plants and *LOX2/3*-deficient plants than between WT plants and *LOX2/3*-deficient plants, indicating a weak, positive effect of *TPS10* expression on apparent plant health. Interestingly, *lox2/3* plants at the edges of monocultures appeared less healthy than WT edge plants, but *lox2/3* plants at the edges of *lox2/3* + *lox2/3xTPS10* mixed cultures did not, indicating a positive neighbor effect of *TPS10* expression on the health of plants in *LOX2/3*-deficient populations.

## *TPS10*-expressing plants increase the infestation rates of their neighbors by the stem-boring weevil *Trichobaris mucorea*

At the end of season two, we found that 35% of all surviving plants were infested with *T. mucorea* larvae. An adult *T. mucorea* is pictured in *Figure 10A*. Interestingly, *TPS10* expression was associated with slightly higher infestation rates: 29% of *lox2/3* and 34% of WT plants vs 39% of *TPS10* and *lox2/3xTPS10* plants. Looking at infestation rates by population type, we discovered a small difference between WT (34% infested) and *TPS10* plants (39% infested) which masked a much larger effect: WT plants in WT + *TPS10* mixed populations (55% infested) had an infestation rate more than twice that of WT plants in monocultures (19% infested, corrected p=0.015, *Figure 10B*). We observed the same tendency in *lox2/3* and *lox2/3xTPS10* populations: 33% of *lox2/3* plants in mixed populations with *lox2/3xTPS10* plants were infested, vs only 16% of *lox2/3* plants in monocultures; interestingly, *lox2/3xTPS10* plants tended to have higher infestation rates than *lox2/3* plants in populations (*Figure 10C*). Likely due to higher plant mortality in *lox2/3* populations (*Figure 9*), replicate numbers were too low for these effects to be significant (corrected p-values =1 for the comparison of infestation rates of *lox2/3* plants in mono- vs mixed culture, and of *lox2/3* vs *lox2/3xTPS10* monocultures). Likely also due to low replicate numbers (n = 7–10), there were no significant differences in infestation among individuals (corrected p-values = 1 in pairwise tests), but 30% of *TPS10*, 38% of WT, 50% of *lox2/3*, and 57% of *lox2/3xTPS10* individuals were infested. Based on these trends, had there been a significant effect for individuals, it would have been due to *LOX2/3* deficiency and not *TPS10* expression.

Thus, although *TPS10* expression did not significantly alter plants' rates of infestation by *T. mucorea*, it did significantly increase the infestation of neighboring plants.

## *TPS10* expression and *LOX2/3* deficiency interact to alter the distribution of *Geocoris* spp. activity in populations

Generalist *Geocoris* spp. predators co-occur with *N. attenuata* plants, and prey on their herbivores in response to specific HIPVs including GLVs and TAB, in an example of indirect defense (*Kessler and Baldwin, 2001*; *Allmann and Baldwin, 2010*; *Schuman et al., 2012*; *2013*). Near the end of experiments in season two, *Geocoris pallens* were abundant and *Geocoris punctipes* were also present at the field site. We considered that plants in populations with *TPS10*-expressing neighbors might either benefit from enhanced attraction of *Geocoris* spp., or suffer in competition for the predation services of *Geocoris* spp. We also reasoned that the lack of all other HIPVs in *LOX2/3*-deficient plants could alter the strength and direction of such a relationship.

To test the overarching hypothesis that *TPS10*-expressing plants in populations alter the distribution of predation by *Geocoris* spp., we conducted predation activity assays from June 21st to June 27th on edge and center plants that were similar in size and apparent health. We treated LOX2/3-expressing (WT, TPS10) and *LOX2/3*-deficient (*lox2/3*, *lox2/3xTPS10*) populations as two separate experimental groups because it was not possible to match plants across these two groups for apparent health or replicate numbers (both were lower for *lox2/3* and *lox2/3xTPS10* populations). Selected *lox2/3* and *lox2/3xTPS10* populations had a mode of 3 remaining plants, range 2–5; and

selected WT and *TPS10* populations had a mode of 5 remaining plants, range 3–5; within these two groups, there was no significant difference between mono- and mixed cultures in number of plants remaining (G-tests, corrected p>0.1). Plants were baited with *M. sexta* eggs and larvae (*Figure 11A*), and we counted the numbers predated by *Geocoris* spp. over 24 hr (source data file for *Figure 11*, *Kessler and Baldwin, 2001*).

*TPS10*-expressing plants significantly altered predation rates at the edges and centers of populations, and whether the effect was at the edge or the center depended on *LOX2/3* expression (*Figure 11B,C*). Specifically, *TPS10* expression increased predation from edge plants in *lox2/3* and *lox2/3xTPS10* populations (Fisher's exact tests, corrected p-values<0.05) but not in WT and *TPS10* populations (21–34%, corrected p-values >0.3) (*Figure 11B*). At the edges of monocultures, *lox2/3xTPS10* plants received 43% predation whereas *lox2/3* plants received only 16% (corrected p=0.049), and *lox2/3* plants at the edges of *lox2/3* + *lox2/3xTPS10* mixed cultures were intermediate with 23% (*lox2/3* mono v. *lox2/3* + *lox2/3xTPS10* mixed, corrected p=0.723, *lox2/3xTPS10* mono v. *lox2/3* + *lox2/3xTPS10* mixed, corrected p=0.457). On the other hand, *TPS10* expression decreased, rather than increased predation from center plants in WT + *TPS10* populations (from 46% to 12%, corrected p<0.04) and had no effect on center plants in *LOX2/3*-deficient populations (ranging from 13% to 23%, corrected p-values=1, *Figure 11C*). WT plants received 46% predation at the center of monocultures, while *TPS10* plants received 38% predation at the center of mixed cultures and only 12% at the center of monocultures (WT mono v. WT + *TPS10* mixed, corrected p=0.618; WT mono v. *TPS10* mono, corrected p=0.012; WT + *TPS10* mixed v. *TPS10* mono, corrected p=0.031).

Thus in populations of *lox2/3* and *lox2/3xTPS10* plants deficient in all HIPVs except for the engineered TPS10 volatiles, *TPS10* expression by edge plants increased predator activity on those plants. In contrast, *TPS10* expression by center plants failed to attract more predator activity to the centers of populations. In populations of WT and *TPS10* plants with intact WT HIPVs, *TPS10* expression did not increase predator activity at all; however, *TPS10* expression by edge plants seemed to reduce the penetration of predators to center plants. Thus, depending the richness of the plants' HIPV blends, *TPS10* expression had either positive direct effects on predator activity for emitting plants, or negative indirect effects for their neighbors.

## Discussion

In this study, we sought to fulfil the four postulates of community genetics (*Wymore et al., 2011*; *Whitham et al., 2012*) to uncover the community effects of genes controlling direct and indirect defenses in the wild tobacco *N. attenuata*. *N. attenuata* has already been shown to fulfill postulate one: significantly affecting an ecological community of other plants and animals, and native microbes (reviewed in *Schuman and Baldwin, 2012*; see also *Long et al., 2010*; *Meldau et al., 2012*; *Machado et al., 2013*; *Santhanam et al., 2014*). Postulate two, demonstration of genetically based traits, is ensured by our method of altering the expression of specific genes: *LOX2*, *LOX3*, and *TPS10* (*Figure 1* and associated source data files, Appendices 1, 2). Thus, we created transgenic lines from a single genotype of *N. attenuata* and used them to investigate both individual- and population-level effects of two HIPVs, (E)-α-bergamotene (TAB) and (E)-β-farnesene (TBF), produced by the sesquiterpene synthase TPS10, in plants which either had wild-type direct defenses and HIPVs, or reduced production of direct defenses and other HIPVs. We used these plants to evaluate postulate three: different effects on community processes of conspecifics differing in *LOX2/3* and *TPS10* expression, and postulate four: predictable effects on those community processes when the genes are manipulated.

Specifically, we found that *LOX2/3*-mediated defense had a strong and direct effect on plant resistance to foliar herbivores and plant mortality (*Figures 4*, *5* and associated source data files, *Figure 9*), as predicted (*Kessler et al., 2004*). Interestingly, the effect of *LOX2/3* expression on herbivore abundance was less clear: *LOX2/3* deficiency was associated with significantly increased herbivore abundance for edge plants in populations, but not overall (*Figure 6*). Furthermore, although *TPS10* expression did not significantly affect foliar herbivore damage or performance, it did consistently reduce foliar herbivore abundance (*Figure 6*). It should be noted that the herbivore damage data in *Figure 5B* were collected just over 2 weeks earlier than the herbivore abundance data in *Figure 6*; however, the relative amounts of damage on plants at the time that herbivore abundance data were collected still reflected the patterns shown in *Figure 5*, with LOX2/3-deficient

genotypes having visibly more damage and WT and *TPS10* plants not differing visibly in herbivore damage levels.

More surprisingly, we found population-level effects of *TPS10* volatiles which were as great as the individual-level effects of total *LOX2/3*-mediated defenses, and dependent on *LOX2/3* expression. Single *TPS10*-expressing plants reduced the mortality of plants in *LOX2/3*-deficient populations (*Figure 9*), corresponding to an increase in apparent health of those plants. Additionally, *TPS10* expression significantly increased the infestation rate of neighbors by the stem-boring weevil *T. mucorea* (*Figure 10*), and *TPS10* expression in a population strongly altered the distribution of predation services among edge and center plants, depending on *LOX2/3*-mediated defense (*Figure 11* and associated source data file).

It should be noted that our experimental design did not support the analysis of population-level effects resulting from changing *LOX2* and *LOX3* expression in single members of a population. Had we included these treatment groups as well (e.g., populations with a central *lox2/3* plant surrounded by WT, and with a central *lox2/3xTPS10* plant surrounded by *TPS10*), we predict that we would also have observed significant neighbor effects of LOX2/3 expression in populations. However, given the magnitude of direct effects of *LOX2/3* expression, it would be surprising if the indirect population-level effects from manipulating *LOX2/3* expression in single plants were larger than the direct effects—as was generally observed from the manipulation of *TPS10* expression. Our experiment was designed to determine whether TPS10 products have direct and neighbor effects in the presence, or absence, of total HIPVs and jasmonate-mediated defense.

We found no effects on non-target metabolites, (*Figure 1—source data 3*, Appendix 2), and only minor effects on plant size and growth which were all associated with the alleviation of the costs of *LOX2/3*-mediated defense for single plants, benefits which disappeared under high herbivory loads (*Figures 7*, *8* and associated source data file, *Schuman et al., 2014*).

## Plant productivity depends on herbivore damage rates, LOX2/3 and TPS10 expression

It is well known that jasmonate-mediated defenses can be costly in the absence of herbivores (reviewed in e.g., *Steppuhn and Baldwin, 2008*), and under low levels of damage from herbivores in season one (*Figure 5A*), growth and early flower production was greater in *lox2/3* and *lox2/3xTPS10* than in WT and *TPS10* plants (*Figures 7*, *8* and associated source data file). These data indicate that our measurements were sufficiently sensitive to detect developmental costs of defense metabolite production. Attempts to engineer the emission of terpene volatiles have sometimes resulted in plants with stunted growth or symptoms of autotoxicity (*Robert et al., 2013*; reviewed in *Dudareva and Pichersky, 2008*; *Dudareva et al., 2013*). However, we did not detect any consistent costs of *TPS10* overexpression: *TPS10* plants were generally not smaller than WT (*Figure 8* and associated source data file). The same line of TPS10 also grew similarly to WT in a greenhouse competition study (line 10–3 in *Schuman et al., 2014*).

However, there were effects of *TPS10* expression on flowering which differed depending on plants' position in different populations. In particular, under high herbivory in season two, *TPS10* expression strongly decreased flower production for well-defended individual plants, but promoted flower production in *LOX2/3*-deficient individuals (*Figure 8B*). In contrast, flower production was not affected by *TPS10* expression for edge plants in populations, and was slightly reduced for *TPS10*-expressing plants in the centers of populations (*Figure 8D,F*). For central plants, in contrast to individuals, the reduction in flower production due to *TPS10* expression was similar both for *LOX2/3*-expressing and *LOX2/3*-deficient plants. Overall, these data indicate that *TPS10* expression may have altered the relative attractiveness or apparency of *TPS10* plants, with results depending strongly on plants' neighbors. The potential mechanisms in terms of herbivore abundance and predation rates, and consequences in terms of health and mortality, are discussed further below.

Higher levels of damage from foliar herbivores in season two (*Figure 5B*) were likely the main factor which shifted *LOX2/3*-mediated defenses from costly, in terms of early flower production (*Figure 7*), to beneficial, in terms of reduced plant mortality before and during the reproductive phase, and increased apparent plant health (*Figure 9* and *Figure 9—figure supplement 1*). Overall, there were few differences in growth and size among genotypes (*Figure 8* and associated source data file, and results reported in text), making it reasonable to attribute changes in herbivore damage,

mortality, and other interactions to the manipulated chemical traits, rather than plant apparency or attractiveness based on size and development.

## *LOX2/3* expression has large direct effects on plants, while *TPS10* expression has large indirect effects

*LOX2/3* deficiency significantly increased damage from foliar herbivores and growth of the specialist *M. sexta*, whereas *TPS10* expression had no effect on either interaction (*Figures 6*, *7*). There was, furthermore, no effect of plant population type (mixed or monoculture) on damage from foliar herbivores in field experiments (*Figure 5* and associated source data files). Together, these data indicate that feeding of *M. sexta* larvae and other naturally occurring foliar herbivores are not directly affected by *TPS10* volatiles. However, *TPS10* volatiles may have altered the attraction of herbivores to plants: TPS10-expressing plants had consistently smaller populations of foliar herbivores by the end of season two, despite the fact that these plants tended to look healthier, and independently of *LOX2/3* expression (*Figure 6*). In contrast, ovipositing *T. mucorea* mothers may be attracted to *TPS10* volatiles (*Figure 10*). Interestingly, *M. sexta* moths laid fewer eggs on plants for which the headspace had been supplemented with TAB (*Kessler and Baldwin, 2001*). A direct effect on colonizing herbivores and ovipositing females may result in indirect effects in terms of associational resistance or associational susceptibility for neighboring plants (*Kessler and Baldwin, 2001*; *Barbosa et al., 2009*). Although such indirect effects were not detected in foliar herbivore abundance on neighboring WT and *lox2/3* plants (*Figure 6C*), *TPS10* expression did significantly affect neighbors' mortality (*Figure 9*), apparent health, infestation by *T. mucorea* larvae (*Figure 10*), and predation services (*Figure 11*).

The most straightforward indirect effects of *TPS10* expression were evident in open headspace trappings (*Figure 3*): the presence of a *lox2/3xTPS10* plant in the center of a population was associated with TAB in the headspace of neighboring *lox2/3* plants, and TAB was not found in the open headspaces of *lox2/3* individuals or monocultures. The most parsimonious explanations are diffusion of volatiles into neighbors' headspace, or adherence and re-release of volatiles from the surface of neighbor plants, as has been shown in birch (*Betula* spp.) by *Himanen et al. (2010)*. Because no TAB was detectable in extracts from the leaves of *lox2/3* plants even when plants were much larger and more likely to have direct contact with *TPS10* neighbors, it is unlikely that the TAB detected in open trappings of *lox2/3* headspaces came from the stimulation of TAB biosynthesis in *lox2/3* (*Figure 1C*, *Figure 1—source data 3*).

Indirect effects of *TPS10* volatiles on neighboring plants may be attributed to the volatile compounds themselves, because we found no evidence that *TPS10* volatiles affected the defenses of emitting plants, or of neighbor plants in populations (Appendices 1, 2, *Figure 1* and associated source data files); and *TPS10* plants also did not alter the defense response of neighbors in a glasshouse competition experiment (*Schuman et al., 2014*). In contrast, due to the large suite of defenses regulated by *LOX2* and *LOX3* products, it would have been more difficult to determine the proximate cause of neighbor effects resulting from manipulation of *LOX2* and *LOX3* expression. Although several components of plant HIPV blends have been shown to prime the direct and indirect defenses of neighbors in different systems (reviewed in *Baldwin et al., 2006*; *Arimura et al., 2010*; see *Heil and Kost, 2006*), GLVs may often be the active cues (*Engelberth et al., 2004*; *Kessler et al., 2006*; *Frost et al., 2008*). We did not mix GLV-deficient with GLV-emitting plants in our experimental design, but we can conclude that *TPS10* volatiles had no detectable direct effects on neighbor plants in our set-up.

## Indirect effects of *TPS10* HIPVs may benefit or harm emitters, depending on the emission and *LOX2/3*-mediated defense levels of their neighbors

We observed multiple significant indirect effects of *TPS10* expression in populations, as mentioned above; and two of these were strongest in mixed cultures containing only one *TPS10*-expressing plant. The mortality of *lox2/3* and *lox2/3xTPS10* plants in monocultures was significantly higher than that of WT and *TPS10* plants in monocultures, but the mortality of *lox2/3* + *lox2/3xTPS10* plants in mixed cultures was not higher than that of WT + *TPS10* plants in mixed cultures (*Figure 9*). Thus, a single *TPS10* emitter in populations of otherwise defenseless plants reduced overall mortality to

levels comparable to well-defended populations containing single *TPS10* emitters. This apparent protective effect of *TPS10* volatiles was weaker for monocultures of *lox2/3xTPS10* plants. This may be because *TPS10* volatile emission makes plants more apparent to herbivores like *T. mucorea* as well as beneficial insects like *Geocoris* spp. (**Figures 9**, **10**), and if plants are otherwise poorly defended, greater apparency may be dangerous (*Feeny, 1976*). Interestingly, although there were no statistically significant differences in mortality among populations of WT and *TPS10* plants, mortality was lowest in *TPS10* monocultures; and WT + *TPS10* mixed cultures had mortality rates 1.5-fold that of *TPS10* monocultures (**Figure 9**). This indicates that well-defended *TPS10* plants may have benefited directly from emitting *TPS10* volatiles, and may have additionally increased the mortality of non-emitting neighbors. If so, *TPS10* volatiles may increase plants' competitive ability by increasing neighbor mortality.

Indeed, *TPS10* plants had a dramatic effect on the infestation of their WT neighbors by *T. mucorea* weevils (**Figure 10**). *TPS10* volatiles seemed only to affect the infestation rates of plant populations; for individual plants, jasmonate-mediated defense is likely more important (*Diezel et al., 2011*, and see 'Results'). Infestation patterns in populations (**Figure 10**) strongly indicate that ovipositing *T. mucorea* adults seek populations of plants based on their volatile emission and then oviposit on all plants in the population. Infestation rates did tend to be higher for *lox2/3xTPS10* plants than for *lox2/3* plants in populations (**Figure 10C**), indicating that the absence of other volatiles such as GLVs may reduce plants' apparency to ovipositing *T. mucorea*.

It is interesting that the *TPS10*-mediated increase in *T. mucorea* infestation was greater for WT plants in mixed WT + *TPS10* populations than for *TPS10* plants in monocultures (**Figure 10**): though *TPS10* monocultures tended to have higher infestation rates than WT monocultures, this difference was not statistically significant, whereas the difference between WT plants in monocultures and WT plants in mixed cultures was both larger and significant. It is tempting to speculate that *T. mucorea* may be attracted to lower relative abundance of *TPS10* volatiles, and deterred by higher relative abundance of these volatiles in the total mixture of the plant headspace. If true, this could provide a mechanism by which *TPS10* emitting plants could inflict an indirect cost of emission on non-emitting neighbors, which would be always slightly higher than their own cost of emission. Furthermore, if attraction or deterrence depended on the ratio of *TPS10* volatiles to other plant volatiles, the effects might differ for *TPS10* and *lox2/3xTPS10* plants (*Bruce et al., 2005*).

The infliction of such an indirect cost on neighbors is reminiscent of extortionist strategies in the iterated prisoner's dilemma of evolutionary game theory: extortionist strategies lock a player's payoff to its opponent's payoff plus some positive number, thereby ensuring that the extortionist always benefits compared to its opponent (*Press and Dyson, 2012*). Interestingly, such strategies have been shown in simulations not to be evolutionarily stable, but to catalyze the evolution of cooperation in populations (*Hilbe et al., 2013*). Of course it is possible that the manipulation of *LOX2* and *LOX3* expression within populations would also reveal such extortionist-like effects, for example if *T. mucorea* preferred to oviposit in *TPS10* plants over *lox2/3xTPS10* plants in the same population, which would support the hypothesis that *T. mucorea* preference reflects the ratio between *TPS10* volatiles and other HIPVs.

## Effects of *TPS10* volatiles on indirect defense depend on *LOX2/3*-mediated direct and indirect defense

Predation rates of *M. sexta* herbivores from edge and center plants in populations (**Figure 11**) indicate that populations' total HIPV emissions can alter the roles of specific volatiles for indirect defense. In *lox2/3* + *lox2/3xTPS10* populations, deficient in all HIPVs except those provided by *TPS10* expression, *lox2/3xTPS10* edge plants of monocultures experienced higher predation rates, whereas predation rates were similar for center *lox2/3xTPS10* and *lox2/3* plants in mono- and mixed cultures. For *TPS10* and WT plants, effects on predation rates were exactly the opposite. There was significantly less predation of *M. sexta* from *TPS10* center plants in monocultures vs mixed cultures, but *TPS10* edge plants had predation rates similar to WT. Together, these results indicate that the effect of *TPS10* volatiles on indirect defense was positive for *lox2/3xTPS10* plants, but negative for *TPS10* plants, and only significant for plants in monocultures. In mixed cultures, *TPS10* volatiles had no effect on the predation of *M. sexta* herbivores from emitting plants or their neighbors. It should be noted that due to higher mortality in *lox2/3* + *lox2/3xTPS10* populations, center plants were sometimes more exposed in these populations as a result of edge plants dying. However, the fact

that *TPS10* volatiles benefitted only edge plants in these populations indicate that *Geocoris* spp. predators nevertheless distinguished between edge and center plants.

We conclude that *TPS10* expression had different consequences for plants' indirect defense, depending on whether one or all plants in a population expressed *TPS10*, and whether those plants also emitted other HIPVs. If only one plant in the population expressed *TPS10*, or if other HIPVs were also emitted in the population, *TPS10* expression did not enhance indirect defense and was sometimes detrimental to emitters. The attraction of most, or all, predators and parasitoids to HIPVs depends on learned associations between HIPVs and prey: naïve predators are just as likely to respond as they are not to respond (*Allison and Hare, 2009*). Because *LOX2/3*-deficient plants were heavily attacked by herbivores (*Figure 5*), predators could associate *TPS10* volatiles emitted by *lox2/3xTPS10* plants with herbivore damage (though damage did not necessarily reflect abundance—see *Figure 6*). Because *lox2/3* and *lox2/3xTPS10* plants had reduced emissions of other HIPVs (*Figure 1* and associated source data files, Appendix 2), there were few other volatiles which could provide the basis for learned associations between HIPVs and prey. On the other hand, TPS10 volatiles emitted constitutively by less heavily attacked *TPS10* plants (*Figure 5*) would be less often associated with feeding herbivores than the endogenous HIPVs released only when herbivores attack, and all the more so if *TPS10*-expressing plants supported fewer foliar herbivores (*Figure 6*). Perhaps because 'honest' endogenous HIPVs were also emitted by WT plants, *TPS10* plants had no advantage over WT in attracting predators.

The concept of honest vs deceptive, or 'cry wolf' HIPV emission has been investigated in the context of emission proportionate, or disproportionate to numbers of attacking herbivores (*Shiojiri et al., 2010*). But what happens when plants 'cry wolf' by constitutively emitting volatiles which, in nature, are only associated with herbivore damage? Plants engineered to constitutively emit particular HIPVs can attract more parasitoids when parasitoids are trained to associate engineered volatiles with their prey (*Kappers et al., 2005*; *Schnee et al., 2006*), or when all emitting plants are also experimentally infested (*Rasmann et al., 2005*; *Degenhardt et al., 2009*). Such studies demonstrate that genetic engineering of HIPVs can enhance indirect defense, but they do not demonstrate that constitutive emission is an effective strategy in the complex environments of agriculture, or nature (*Kos et al., 2009*; *Schuman et al., 2012*; *Xiao et al., 2012*). From an arthropod's perspective, constitutive HIPV emission is oxymoronic. If there is no effort to artificially associate constitutively emitted volatiles with prey, predators, and parasitoids may not learn to respond, or worse, learn to actively ignore the volatile emissions. Whether achieved via genetic methods or synthetic release stations (*Kaplan, 2012*), constitutive emission is unlikely to enhance indirect defense if prey densities are low, or if plants emit other 'honest' HIPVs.

Moreover, assigning an indirect defense function to HIPVs based on their ability to attract predators or parasitoids to single plants may be misleading. In real populations, the effectiveness of HIPVs likely depends on the density of HIPV emission and the location of emitting plants (*Xiao et al., 2012*). There may be competition among plants in populations either to attract, or to arrest predators and parasitoids of herbivores. Specifically, whether plants benefit from emitting more or less than their neighbors likely depends on the basal level of HIPV emission in a population (low or high), and the location of plants in a population (edge or center). Perhaps, wild-type plants are capable of adjusting their own volatile emission in response to the emission of their neighbors in order to adapt (*Choh et al., 2004*; *Baldwin et al., 2006*; *Yan and Wang, 2006*). In this context, it would be interesting to know whether central *lox2/3* plants surrounded by WT edge plants would experience a similar predation disadvantage as central *TPS10* plants surrounded by WT edge plants, and whether central WT plants surrounded by *lox2/3* edge plants would have an advantage over their neighbors in attracting predators.

## Conclusion

Now that careful and extensive research has established the importance of biodiversity to ecosystem function, the next step is to understand how the two are linked in order to craft informed socioeconomic and environmental policy (*Midgley, 2012*). The community genetics perspective is essential in order to understand biodiversity effects at the level of individual heritable traits (*Whitham et al., 2006*). This is especially true in light of recent research showing that intraspecific, genetic biodiversity is at least as important as interspecific biodiversity for the structure of ecological communities (*Crutsinger et al., 2006*; *Johnson et al., 2006*; *Cook-Patton et al., 2011*; *Moreira and Mooney,*

*2013*) and the recognition that interspecific biodiversity is an emergent property of genetic biodiversity (*Johnson and Stinchcombe, 2007*; *Haloin and Strauss, 2008*).

The work presented here demonstrates that the population-level effects of an indirect defense gene may be greater than its effects on individual plants, demonstrating that the population perspective is vital to our understanding of gene function. Furthermore, our results indicate that populations comprising individuals differing in only a single genetic trait can functionally imitate more diverse communities. Engineering functional diversity may become an important strategy in a world with over 10 billion humans (*United Nations, Department of Economic and Social Affairs, Population Division, 2013*), where every patch of arable land may be planted with custom-engineered high-yield crop monocultures.

## Materials and methods

### WT and transgenic plant lines and glasshouse growth conditions

Seed germination, glasshouse growth conditions, and the *Agrobacterium tumefaciens* (strain LBA 4404)–mediated transformation procedure have been described previously (*Krügel et al., 2002*; *Schuman et al., 2014*). Seeds of the 31st generation of the inbred 'UT' line of *N. attenuata* (Torr. ex S Watson) described by *Krügel et al. (2002)* were used for the wild-type (WT) plants in all experiments. We used homozygous transformed lines of the second transformed generation (T$_2$), each with a single transgene insertion, to either ectopically overexpress *Zea mays TERPENE SYNTHASE 10* (*TPS10*) under the control of a 35S promoter in the pSOL9 plasmid (*TPS10* line A-09-389-6/10-3, described in *Schuman et al., 2014*), or silence *N. attenuata LIPOXYGENASE 2* and *LIPOXYGENASE 3* (*LOX2/3*) via an inverted repeat construct specifically targeting both genes in the pSOL8 plasmid (line A-07-707-2, sequences targeting *NaLOX2* and *NaLOX3* described in *Allmann et al. (2010)* (*Figure 1*). Vector construction and the pSOL8 and pSOL9 plasmids have been described previously (*Gase et al., 2011*). To combine the ectopic overexpression of *TPS10* with the silencing of *LOX2/3*, a hemizygous cross was created between the *lox2/3* and *TPS10* T2 homozygous lines (*lox2/3xTPS10*, *Figure 1*). Further details of transgenic line screening and characterization are given in Appendix 3.

### Field plantations and growth conditions

For field experiments, seedlings were transferred to 50 mm peat pellets (Jiffy) 15 days after germination and gradually hardened to the environmental conditions of high sunlight and low relative humidity over 10 days. Small, adapted, size-matched rosette-stage plants were transplanted into a field plot in a native habitat in Utah located at latitude 37.146, longitude 114.020. Field plantations were conducted under APHIS permission numbers 06-242-3r-a3, 10-349-102r, and 11-350-101r. Plants were watered thoroughly once at planting and as needed over the first 2 weeks until roots were established; all plants received the same watering regime in each year. WT, *TPS10*, *lox2/3*, and *lox2/3xTPS10* plants were arranged as individuals and populations according to the descriptions in *Figure 2* and *Figure 2—figure supplement 2* (n = 12 in season one, n = 15 in season two of each individual or population type).

A distance of 0.5 m is sufficient for *N. attenuata*'s herbivores and their predators to distinguish odor from individual plants (*Kessler and Baldwin, 2001*; *Schuman et al., 2012*): so that insects could perceive plants in a population as a single unit, they were planted in 0.4 × 0.4 m squares, while the distance between single individuals/populations was ca. 1.5 m. Individual and population types were marked with small (ca. 5 cm long, Ø 0.5 cm) bamboo sticks having different numbers of horizontal markings, which were checked during experiments only when establishing treatment groups, or confirming current location within the experiment. During experiments and data collection, plants were referred to only by their number or position (e.g., n1 population 2 upper left, where upper left was defined from a standardized observer position). Plant genotypes were never identified during data collection to avoid bias, and experimental treatments and data collection were spatially and temporally randomized corresponding to the randomized arrangement of individuals and populations in the field plot (*Figure 2—figure supplement 2*).

## Tissue samples from glasshouse- and field-grown plants

Prior to measuring gene transcripts and induced defenses or HIPVs, glasshouse-grown plants, which were not damaged by herbivores, were treated with wounding and *M. sexta* oral secretions (W + OS) as a standardized method to mimic herbivore feeding as described previously (*Schuman et al., 2012*). The first fully-expanded leaf (node +1) was used for quantifying gene transcript (*LOX2*, *LOX3*, *TPS10*) and jasmonate (JA, JA-Ile) accumulation on a set of bolting plants. The adjacent older leaf (node +2) was used to measure headspace volatiles on the same plants (data in Appendix 2, and see 'Quantification of volatiles in the headspace of glasshouse-grown plants'). A separate set of plants was used following the same procedure for WT and *TPS10* jasmonates and GLVs, and the results are reported in *Schuman et al. (2014)* and ratios to WT are shown in *Figure 1B*. Control plants were left untreated. Treated leaves, and leaves at the same position on control plants were cut at the petiole and wrapped in a double layer of aluminum foil, then immediately flash-frozen in liquid nitrogen, and kept at −80°C until processing and analysis.

For field-grown plants, five similar, mature, non-senescent stem leaves were harvested after the end of predation assays, on June 28th of experimental season two. Two of these leaves had been damaged by the *M. sexta* first-instar larvae used for predation assays, and three were systemic leaves with similar levels of damage from other naturally-occurring herbivores. Leaves were cut at the petiole, pooled in a single sample per plant and wrapped in a double layer of aluminum foil, then immediately frozen on dry ice insulated with ice packs frozen at −20°C. Samples were stored at −20°C until transport to the MPICE on dry ice, where they were kept at −80°C until processing and analysis.

All sample processing was carried out over liquid nitrogen until the addition of the extraction solvents. Prior to analysis, entire leaves were crushed with a mortar and pestle and each transferred to a 2 mL microcentrifuge tube for storage. For specific measurements, aliquots were weighed into microcentrifuge tubes containing two steel balls and finely ground in a GenoGrinder (SPEX Certi Prep, Metuchen, NJ) prior to extraction.

## Quantification of JA and JA-Ile in leaf tissue

JA and the active jasmonate hormone JA-Ile were extracted from 100 mg of leaf tissue (from n = 4 plants) using ethyl acetate spiked with internal standards (IS) and quantified by LC-MS/MS (Varian, Palo Alto, CA) as described by *Oh et al. (2012)*, with the modification that 100 ng rather than 200 ng of $[^2H_2]$JA, and 20 ng rather than 40 ng of JA-$[^{13}C_6]$Ile were used as internal standards (IS). Individual jasmonates were quantified in ng by comparison to the corresponding IS peak area and normalized per g leaf tissue fresh mass (FM). The quantification of TPI in field tissue samples as an indicator of jasmonate signaling is described in Appendix 3.

## Quantification of volatiles in the headspace of glasshouse-grown plants

For GLVs, the +2 leaf (n = 4 plants) was enclosed immediately after W + OS elicitation in two 50 mL PET cups (Huhtamaki, Finland) lined on the edges with foam to protect leaves and with an activated charcoal filter attached to one side for incoming air, and secured with miniature claw-style hair clips as described previously (*Schuman et al., 2009*). For *TPS10* products in the headspace of glasshouse-grown plants (n = 4), leaves were enclosed 24 hr after W + OS treatment and the headspace was collected from 24–32 h during the peak emission of sesquiterpenes in *N. attenuata* (*Halitschke et al., 2000*). Headspace VOCs were collected on 20 mg of Poropak Q (*Tholl et al., 2006*; Sigma–Aldrich, St. Louis, MO) in self-packed filters (bodies and materials from ARS Inc., Gainesville, FL) by drawing ambient air through these clip cages at 300 mL min$^{-1}$ using a manifold with screw-close valves set to provide equal outflow, via pushing air at 2 to 3 bar through a Venturi aspirator as described previously (*Oh et al., 2012*). Background VOCs present in ambient air were collected using empty trapping containers and background signals were later subtracted if necessary from raw intensities of plant samples prior to further processing.

After trapping, Porapak Q filters were stored at −20°C until extraction by addition of 320 ng of tetralin as an internal standard (IS), and elution of volatiles with 250 mL of dichloromethane (Sigma–Aldrich). Filters were eluted into a GC vial containing a 250 µL glass insert. Samples were analyzed on one of two different GC–MS instruments from Varian with columns from Phenomenex (Torrance, CA; 30 m × 0.25 mm i.d., 0.25 µm film thickness). 1 µL of each sample was injected in splitless

mode, and then the injectors were returned to a 1:70 split ratio from 2 min after injection through the end of each run. He carrier gas was used with a column flow of 1 mL min$^{-1}$. GLV samples were analyzed by a CP-3800 GC Saturn 2000 ion trap MS with a polar ZB-wax column and a CP-8200 autoinjector; the GC and MS were programmed as previously described for this instrument (*Schuman et al., 2012*), and compounds were separated by a temperature ramp of 5°C min$^{-1}$ between 40°C and 185°C. *TPS10* products and other VOCs were analyzed on a CP-3800 GC coupled to a Saturn 4000 ion-trap mass spectrometer with a nonpolar ZB5 column and a CP-8400 autoinjector; The GC and MS were programmed as previously described for this instrument (*Oh et al., 2012*), and compounds were separated by a temperature ramp of 5°C min$^{-1}$ between 40°C and 180°C.

Individual volatile compound peaks were quantified using the combined peak area of two specific and abundant ion traces per compound using MS Work Station Data Analysis software (Varian) and normalized by the 104 + 132 ion trace peak area from tetralin in each sample. The identification of compounds was conducted by comparing GC retention times and mass spectra to those of standards and mass spectra databases: Wiley version 6 and NIST (National Institute of Standards and Technology) spectral libraries.

The area of trapped leaves was quantified for comparison by scanning and calculating areas in pixels using SigmaScan (Systat Software Inc., San Jose, CA), and subsequently converting pixels to cm$^2$ using a size standard which was scanned with leaves.

## Quantification of plant volatiles in hexane extracts of field tissue samples

Plant volatiles were quantified in tissue from field-grown plants (n = 9–22). Frozen tissue (100 mg) was spiked with 800 ng tetralin as an internal standard (Sigma–Aldrich) and extracted in 300 μL hexane during an overnight rotating incubation at RT. Tissue was allowed to settle and 100 μL of water- and tissue-free hexane was transferred to a GC vial containing a 250 μL microinsert. Samples were analyzed by a Varian CP-3800 GC-Saturn 4000 ion trap MS connected to a ZB5 column (30 m × 0.25 mm i.d., 0.25 μm film thickness; Phenomenex) as described in 'Quantification of volatiles in the headspace of glasshouse-grown plants'. Analyte quantities were expressed as percent tetralin IS per 100 mg FM.

## Quantification of volatiles in the open headspace of field-grown plants

For the data in *Figure 3*, on May 16th of season two, four activated charcoal filters were placed around each plant (n = 4) as shown in *Figure 3A* and the headspace was not enclosed. The open headspace was sampled for 7 h during the day at which time emission of most volatiles was previously found to be greatest (Appendix 2). After elution (see Appendix 3), eluents from all four filters were combined and concentrated to a volume of 15–20 μL under a gentle stream of nitrogen gas. Pooled and concentrated samples were analyzed by Agilent 6890N GC equipped with an Agilent 7683 auto-injector coupled with a LECO (St. Joseph, MI) Pegasus III time-of-flight MS with a 4D thermal modulator upgrade. Injected samples were separated first on a nonpolar column (C1 RTX-5MS, 20 m × 250 μm i.d. × 0.5 μm; Restek, Bellefonte, PA) and every 6 s (modulation time) transferred to a midpolar column (DB-17, 0.890 m × 100 μm i.d. × 0.1 μm; Agilent Technologies, Santa Clara, CA) for the second separation. Chromatography and analysis conditions as well as deconvolution, alignment, and integration of VOC analyte peaks are described in *Gaquerel et al. (2009)*. During peak table alignment using the comparison feature imbedded in ChromaToF software (LECO), mass spectra alignment was accepted at a similarity threshold of 500/1000. Individual volatile compound peaks were normalized by the peak area of the tetralin IS in each sample. Analyte quantities were expressed as percent tetralin IS per plant.

## Measurement of *M. sexta* growth

As for OS collection, we used *M. sexta* neonates from an in-house colony at the MPICE. One *M. sexta* neonate per plant was placed on the youngest rosette leaf of elongated plants (n = 25 larvae). Larval movement was unrestricted within a plant, but plants were spaced on the table so that larvae could not move directly between plants. Larval mass was determined on days 8 and 12, corresponding on average to the third and fourth larval instars. Plant genotypes were randomized spatially on a

single glasshouse table and temporally in placement and weighing of larvae. Due to mortality and to larval movement off of plants, 11–16 larvae remained within each treatment group by day 12.

### Assessment of foliar herbivore damage

Total canopy damage due to herbivores occurring naturally on the field plot was quantified on June 9th in season one, and on June 2nd in season two. Damage was calculated as described in *Schuman et al. (2012)* by identifying damage from specific herbivores according to their characteristic feeding patterns, counting the number of leaves per plant (small leaves were counted as 1/5 to 1/2 of a leaf based on leaf area and large leaves were counted as 1 leaf), estimating the total percentage of leaf area damage due to each herbivore, and dividing the total leaf area damage from each herbivore by the total number of leaves. Leaf area damage was estimated in categories of 1%, 5%, 10%, 15%, and so on, in steps of 5%. All such damage estimates were made by MCS. Plants with a health index <2.5 (see 'Ranking of apparent plant health') were excluded from herbivore damage calculations as it was difficult to accurately assess herbivore damage to these plants.

### Herbivore and predator counts

In the morning of June 26th in season two, a subset of focal plants with a health index ≥2 was chosen (see 'Ranking of apparent plant health') and numbers of herbivorous and predacious arthropods present on the shoots of these plants were counted (total n's in *Figure 6—source data 1*, *2*). The counting was done by a team of two people in parallel moving down the plot in the same direction, so as to complete counts as quickly as possible and to avoid disturbing plants prior to counting. Plants were randomly assigned to each counter. Counting proceeded by looking at each branch of the plant in a systematic order from the top to the bottom of the plant, observing both the adaxial and abaxial sides of leaves as well as full areas of stems. In this way especially mobile arthropods which tend to feed near the top of plants were either counted before they fled (*Epitrix* spp.), or could be counted once they had relocated to the base of the plant (*T. notatus*).

### Measurement of plant size and flower production

Plant size (rosette diameter, stem length, and branching) was measured at three times during the main period of vegetative growth and beginning of reproduction in season one (May 1st–2nd, 8th–11th, and 16th–17th), and after plants had switched from vegetative growth to reproduction in season two (June 4th–6th), at which time stem diameter was also measured. Rosette diameter was measured as the maximum diameter found by gently laying a ruler over the rosette; stem length was measured from the base of the stem to the tip of the apical inflorescence by placing a ruler beside the stem resting at the base of the rosette; stem diameter was measured by placing a calipers at the base of the stem immediately above the rosette; and all side branches 5 cm or longer were counted.

Additionally, plant reproductive output was monitored by counting the number of flowers (counted once the corolla protruded visibly from the calyx) before removal at two time points after plants began to flower: on May 8th–11th and 16th–17th in season one, and on May 29th and June 4th–6th in season two. Additionally, buds >2 mm long were counted in season two. After these counts, floral meristems were removed as necessary to prevent the distribution of ripe seeds, as is required for field-released transgenic plants.

### Quantification of plant mortality and infestation by *Trichobaris mucorea*

Plant mortality was quantified as the number of plants which died between the end of planting, and the end of the experiment when all plants and remains were removed from the field plot. During removal of plants on June 29th in season two, we split stems open lengthwise to check for *T. mucorea* frass and larvae (*Diezel et al., 2011*). Plants were counted as infested if the main stem was hollow and filled with *T. mucorea* frass, or if a larva was found in such a hollow, frass-filled stem or axial branch. These counts were done by tallying totals without reference to specific plant ID's, and thus could only be analyzed using unreplicated statistical tests.

### Ranking of apparent plant health

In season two, plant health was also ranked by MCS on a scale of 1–5 in increments of 0.5 (1, 1.5, 2, 2.5, ..., 5) using the index in *Figure 7C* of *Schuman et al. (2012)*: 5, healthy, <5% of shoot

senescent, no chlorosis or signs of water stress; 4, 5–10% of shoot senescent and may have visible chlorosis; 3, 10–15% of shoot senescent, some water stress, up to 50% of canopy chlorotic; 2, >15% of shoot senescent, heavy water stress or visibly sick; 1, dead. This scale was sufficiently objective that several observers not directly involved in the experiment arrived independently at the same ranks, to a precision of ±0.5, to describe the health indices of randomly-chosen plants, after training on fewer than 10 example plants.

## Assay of predation by *Geocoris* spp.

Five consecutive predation assays were conducted between June 21st and June 27th at the end of experimental season two, when *Geocoris* spp. and herbivores were abundant. *M. sexta* eggs and larvae (total n's given in the source data file for *Figure 11*) were used as bait for *Geocoris* spp. predators as described by *Kessler and Baldwin (2001)* and *Steppuhn et al. (2008)*. Native *Manduca* spp. did not provide sufficient synchronously oviposited eggs for trials, and so we used *M. sexta* eggs kindly provided by C Miles from her laboratory colony at Binghamton University, Vestal, New York. Size-matched plants with health indices ≥2 in populations were used. We treated LOX2/3-expressing (WT, TPS10) and *LOX2/3*-deficient (*lox2/3*, *lox2/3xTPS10*) populations as two separate experimental groups because it was not possible to match plants across these two groups for apparent health or replicate numbers (both were lower for *lox2/3* and *lox2/3xTPS10* populations). During the five predation assays it was occasionally necessary to select new plants, preventing a repeated measures analysis of the predation data; however, newly selected plants were matched as described. At the end of predation assays, tissue samples were taken from a subset of these plants for the extraction of volatiles and determination of TPI activity (*Figure 1*, *Figure 1—source data 3* and preceding methods).

For each assay, one first-instar *M. sexta* larva and one egg were placed on one lower stem leaf in a standardized position on each plant, and the predation of eggs and larva was monitored after 24 hr. No other *Manduca* spp. eggs or larvae were present on the plants used during the assays. Eggs were affixed to the underside of the leaf with a drop of α-cellulose glue (KVS, Germany), which does not damage plants or affect volatile emission (*Kessler and Baldwin, 2001*). Feeding by the *M. sexta* larva ensured the locally elicited emission of HIPVs. Larvae were considered to be predated when either the larva was missing, but clearly identifiable, fresh *M. sexta* feeding damage was present, or when the predated larval carcass was found on the same plant; eggs were considered predated when the eggshell was empty but intact, except for a small hole which characterizes the typical damage caused by *Geocoris* spp. feeding as described in *Kessler and Baldwin (2001)* and depicted in Figure 3 of *Schuman et al. (2012)*. Eggs occasionally collapse during *Geocoris* spp. predation, but collapsed eggs were not counted unless the eggs were at least 50% empty with a visible feeding hole.

## Statistical analyses

Summary statistics were calculated in Microsoft Excel, as were G-tests, Fisher's exact tests, and tests of Spearman's correlation coefficients, which were calculated using Microsoft Excel spreadsheets from JH Macdonald's online Handbook of Biological Statistics (*Macdonald, 2009*). All other statistical tests were calculated in R version 2.15.2, using the RStudio interface version 0.96.316 or version 0.97.449 (*R Core Team, 2012*). In some cases, data which were clearly not different upon visual inspection and which had been found in similar data sets not to differ significantly were not statistically tested (e.g., rosette diameter in some subsets of data from season one).

When possible, Welch t-tests were used for pairwise comparisons and ANOVAs, linear mixed-effects models, generalized linear models, or generalized linear mixed-effects models were used for multiple comparisons as appropriate; we checked treatment groups graphically (quantile–quantile plots, residual v. fitted plots) and statistically (Shapiro–Wilk test, Bartlett test) for compliance with model assumptions, including normality and homoscedasticity. Data containing too many zero values or not meeting model assumptions were analyzed using ranking tests (Wilcoxon rank sum for pairwise comparisons, Kruskal–Wallis for multiple groups). R package nlme was used for linear mixed-effects models (*Pinheiro et al., 2014*), and packages lme4 (*Bates et al., 2014, ; 2014b*) and multcomp (*Hothorn et al., 2008*) were used for generalized linear mixed-effects models.

The optimal fixed structure for ANOVAs, linear mixed-effects models, generalized linear models, and generalized linear mixed-effects models was found by stepwise model simplification and factor level reduction followed by the comparison of the models via the Akaike information criterion (AIC,

*Akaike, 1973*). Specific examples of model simplification for individual analyses are given in Appendix 3 and *Figure 8—source data 1*.

Holm-Bonferroni p-value corrections were calculated in Microsoft Excel for families of tests on the same data. Like the Bonferroni correction, the Holm-Bonferroni correction controls the familywise error rate and is simple to calculate, but the Holm-Bonferroni method is more powerful (*Holm, 1979*). Briefly, all p-values in a family of tests conducted on the same data set and subsets were listed from smallest to largest. The smallest p-value was multiplied by the total number of tests (n) conducted on that data. If the resulting corrected p-value was <0.05, then the next-smallest p-value was multiplied by n − 1. If the resulting corrected p-value was <0.05, then the third smallest p-value was multiplied by n − 2, and so on. At the first correction resulting in a p-value ≥0.05, that p-value and all larger p-values were considered non-significant. The corrected p-values reported in the text are the products of the correction procedure, for example, (smallest p-value) × n or (second smallest p-value) × (n − 1).

## Acknowledgements

The authors gratefully acknowledge JO Miller for assistance with the processing of open headspace samples and tissue hexane extracts from field experiments; J Koenig, R Ludwig, B Naumann, N Petersen, I Schmidt, and D Schweizer for assistance with glasshouse experiments which required several careful hands and precise timing, and help processing the resulting samples; D Veit for constructing the volatile trapping set-up used in the glasshouse and the 125-sample plate used in the radial diffusion assay of TPI activity; D Kessler for photos of *N. attenuata*, *G. pallens* and *M. sexta* and A Steppuhn for the photo of *T. mucorea*; C miles for providing *M. sexta* eggs and larvae for use in field experiments; and fellow field researchers K Barthel, C Diezel, T Dinh, M Erb, I Galis, J Gershenzon, P Gilardoni, M Heinrich, M Kallenbach, D Kessler, D Marciniak, S Meldau, Y Oh, E Rothe, S Schuck, M Stanton, A Steppke, V Tamhane, L Ullmann-Zeunert, Al Weinhold, Ar Weinhold, and M Woldemariam for insightful scientific discussions and generous assistance when extra hands were needed. We thank APHIS for constructive regulatory oversight and Brigham Young University for the use of Lytle Preserve.

## Additional information

### Author contributions

MCS, Conception and design, Acquisition of data, Analysis and interpretation of data, Drafting or revising the article; SA, Acquisition of data, Analysis and interpretation of data, Drafting or revising the article, Contributed unpublished essential data or reagents; ITB, Conception and design, Acquisition of data, Analysis and interpretation of data, Drafting or revising the article

### Competing interests

ITB: ITB: Senior editor, *eLife.* The other authors declare that no competing interests exist.

### Funding

| Funder | Grant reference number | Author |
|---|---|---|
| Max-Planck-Gesellschaft | | Meredith C Schuman<br>Silke Allmann<br>Ian T Baldwin |
| European Research Council (ERC) | Advanced Grant to ITB 293926 | Meredith C Schuman<br>Ian T Baldwin |
| Nederlandse Organisatie voor Wetenschappelijk Onderzoek | NWO-ALW 821.02.027 | Silke Allmann |
| European Union | FP7-PEOPLE-2011-IEF-302388 | Silke Allmann |

The funders had no role in study design, data collection and interpretation, or the decision to submit the work for publication.

## Author ORCIDs

Meredith C Schuman, http://orcid.org/0000-0003-3159-3534
Ian T Baldwin, http://orcid.org/0000-0001-5371-2974

## Additional files

### Major datasets

The following dataset was generated:

| Author(s) | Year | Dataset title | Dataset URL | Database, license, and accessibility information |
|---|---|---|---|---|
| Schuman MC, All-mann SA, Baldwin IT | 2015 | Data from: Plant defense phenotypes determine the consequences of volatile emission for individuals and neighbors | http://dx.doi.org/10.5061/dryad.qj007 | Available at Dryad Digital Repository under a CC0 Public Domain Dedication. |

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

## Appendix 1

### Herbivore-induced plant defenses and HIPVs are precisely manipulated in lox2/3, TPS10, and lox2/3xTPS10 plants

We confirmed a single insertion of the inverted repeat *LOX2/3* construct in several *lox2/3* lines (*Appendix 1—Figure 1A*) and chose lines with single insertions to screen for reduced jasmonic acid (JA) and jasmonoyl isoleucine (JA-Ile) (*Appendix 1—Figure 1B,C*).

Based on these data, we chose *lox2/3* line 707-2 for further experiments overexpressing *TPS10* plants (line 10–3) also harbor a single transgene insertion (*Schuman et al., 2014*), and both lines were also confirmed to be diploid by flow cytometry. All transgenic lines showed the intended reduction or increase in target gene transcripts: *LOX2* and *LOX3* transcripts accumulated to <7% of WT levels in *lox2/3* and *lox2/3xTPS10*, while *TPS10* transcripts were easily quantifiable in *TPS10* and *lox2/3xTPS10*, but undetectable in WT or *lox2/3* (p-values < 0.01 for pairwise differences in Tukey HSD tests following p-values <0.001 in 1-way ANOVAs on natural log-transformed data, *Appendix 1—Figure 2*). Transcript abundance of *LOX2* and *LOX3* was similar in WT and *TPS10* (p=1 in Tukey HSD tests) and in *lox2/3* and *lox2/3xTPS10* (p=0.471 for *LOX2* and p=1 for *LOX3*); *TPS10* transcript abundance was similar in *TPS10* and *lox2/3xTPS10* (p=0.157).

As a consequence of reduced *LOX3* transcripts (*Allmann et al., 2010*), *lox2/3* and *lox2/3xTPS10* plants accumulated ≤30% of WT JA levels in a glasshouse experiment in which leaves were treated with wounding plus distilled water (W + W) or wounding plus the addition of *Manduca sexta* oral secretions diluted in distilled water (W + OS) to mimic herbivore attack in a precisely timed manner (*Figure 1B*, see also overview of phenotypes in *Figure 1A*). Induced levels of the active jasmonate, JA-Ile, were also reduced to <50% of WT. Absolute values for JA and JA-Ile, and significant pairwise differences in Tukey HSD tests (p-values <0.05) following two-way ANOVAs on natural log-transformed data (p-values <0.002), are shown in *Figure 1—source data 1*. Likewise, reduced *LOX2* transcripts resulted in diminished GLV emissions from *lox2/3* and *lox2/3xTPS10* plants after W + W or W + OS treatment to <2% of WT levels (*Figure 1B*, *Figure 1—source data 1*). Significant pairwise differences in Wilcoxon rank sum tests (Holm-Bonferroni-corrected p-values <0.03) following Kruskal–Wallis tests (corrected p-values ≤0.020) are shown in *Figure 1—source data 1*. In contrast, GLV emission and JA and JA-Ile accumulation were similar in WT and *TPS10* plants (*Figure 1*, *Schuman et al., 2014*), but both *TPS10* and *lox2/3xTPS10* plants emitted elevated amounts of TAB and TBF, more than 10-fold WT levels as measured in a second glasshouse experiment (*Figure 1B*, *Figure 1—source data 2*). Genotypes with the *TPS10* overexpression construct vs those without differed significantly in their emission (Wilcoxon rank sum tests, corrected p-values ≤0.009, shown in *Figure 1—source data 2*).

As in the glasshouse experiments, *lox2/3* and *lox2/3xTPS10* plants in the field had strongly reduced jasmonate-mediated defense and GLV emission, to <4% of WT levels; and *TPS10* and *lox2/3xTPS10* plants emitted elevated levels of TAB and TBF, >18× WT levels (*Figure 1C*; data reported in the following subsection). The reductions in GLVs, JA, and JA-Ile described here for *lox2/3* and *lox2/3xTPS10* are similar to what has been previously reported for homozygous transformed lines silenced either in *LOX2* (GLVs) or *LOX3* (JA, JA-Ile) by separate, specific RNAi constructs (*Allmann et al., 2010*).

In summary, JA, JA-Ile, jasmonate-mediated defense, and GLVs were significantly reduced in *lox2/3* and *lox2/3xTPS10* plants to <4% of WT levels, and *lox2/3xTPS10* and *TPS10* plants produced more than 10 times as much TAB and TBF as WT or *lox2/3* plants; these phenotypes were robust in plants grown under controlled glasshouse conditions and in the field (*Figure 1* and supplements, Appendix 2). This set of transgenic plants (*Figure 1A*) allowed us to manipulate TAB and TBF independently of defense signaling and all other

HIPVs, creating plants with enhanced emission in a well-defended (WT, *TPS10*) or poorly defended (*lox2/3*, *lox2/3xTPS10*) background.

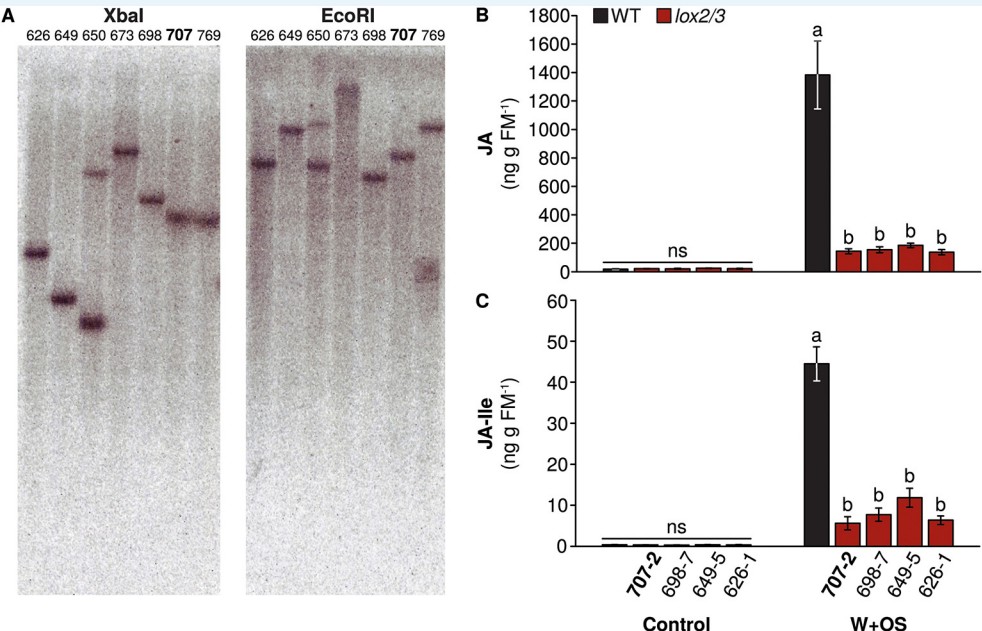

**Appendix 1—Figure 1.** Transgene insertions and reduced jasmonate accumulation in lines of *lox2/3*. (**A**) Southern blot of genomic DNA from multiple lines of *lox2/3* digested with the restriction enzymes XbaI or EcoRI. Line 707 is highlighted: this line was used for experiments. A single transgene insertion in *TPS10* (line 10–3) is shown in **Schuman et al. (2014)**. All lines of *lox2/3* screened had strong reductions in their induced (**B**) JA and (**C**) JA-Ile accumulation (mean ± SEM, n = 6). Line 707-2 was used for further experiments. [a,b] Different letters indicate significant differences (corrected p<0.0001) in Tukey HSD tests following significant (p<0.0001) 1-way ANOVAs on natural log-transformed data (JA: $F_{4,25}$ = 54.15; JA-Ile: $F_{4,25}$ = 18.00); ns, not significant in 1-way ANOVAs (p>0.4).

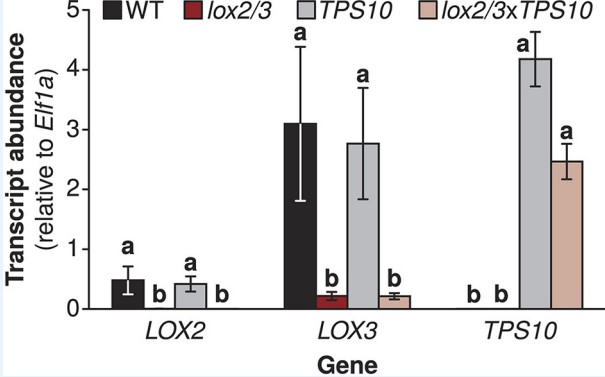

**Appendix 1—Figure 2.** Target gene transcript levels in *lox2/3*, *TPS10*, and *lox2/3xTPS10* plants. Transcript accumulation shows that *lox2/3* plants, *TPS10* plants, and hemizygous crosses (*lox2/3xTPS10*) have the expected silencing or accumulation of the target genes *LOX2*, *LOX3*, and *TPS10* (mean ± SEM, n = 4). [a,b] Different letters indicate significant differences (corrected p<0.01) in Tukey HSD tests following a significant (p<0.001) 1-way ANOVA on natural log-transformed data (*LOX2*: $F_{3,12}$ = 49.49, p<0.001; *LOX3*: $F_{3,12}$ = 13.73, p<0.001; *TPS10*: $F_{3,12}$ = 845.6, p<0.001).

## Appendix 2

### Phenotypic measurements of transgenic lines in experimental field-grown populations.

Trypsin protease inhibitor (TPI) activity, a well-known jasmonate-mediated defense (**van Dam et al., 2001**; **Halitschke and Baldwin, 2003**), is more a stable trait than are JA levels in frozen tissue and thus was used as an indicator of jasmonate-mediated defense levels in field-harvested samples. TPI activity was reduced to <4% of WT activity in *lox2/3* and *lox2/3xTPS10* plants grown in the field. A Kruskal–Wallis test among genotypes indicated a highly significant difference ($\chi^2_3$ = 47.87, corrected p<0.001), as did Wilcoxon rank sum tests between WT and *lox2/3*, and WT and *lox2/3xTPS10* ($W_{24,12}$ = 0, $W_{24,11}$ = 0 and corrected p-values<0.001, **Figure 1C**, **Figure 1—source data 3**). There was no significant difference in TPI activity between WT and *TPS10* ($W_{24,21}$ = 357, corrected p=0.065), as was previously shown in glasshouse experiments (**Schuman et al., 2014**).

GLV pools extracted from leaf tissue samples of field-grown *lox2/3* and *lox2/3xTPS10* plants were <3% of those found in leaf tissue from field-grown WT plants (corrected p-values <0.001 in Wilcoxon rank sum tests), whereas *TPS10* and WT did not differ (corrected p-values = 1 in Wilcoxon rank sum tests, **Figure 1C**). (*E*)-Hexen-2-al and (*Z*)-hexen-3-ol were the only GLVs found in extracts (see also **Schuman et al., 2012**), and both differed significantly by genotype. Significant differences in pairwise comparisons between *lox2/3*, *lox2/3xTPS10*, *TPS10* and WT in Wilcoxon rank sum tests (corrected p<0.001) following Kruskal–Wallis tests (corrected p-values < 0.001) are shown in **Figure 1—source data 3**. There was no significant effect of plant type or population type on extracted GLVs (corrected p-values =1 in Kruskal–Wallis tests across individuals, edge and center plants, or across individuals, mono-, and mixed cultures). Emission of (*Z*)-hexen-3-ol, the only GLV detected from plants having natural background levels of herbivore attack in season one, was also reduced in *lox2/3* and *lox2/3xTPS10* to <16% of WT emission (**Appendix 2—Table 1**). Daytime emission of (*Z*)-hexen-3-ol was low and differed only marginally in a Kruskal–Wallis test among genotypes ($\chi^2_3$ = 9.27, corrected p=0.052), but nighttime emission differed significantly in a Wilcoxon rank sum test between plants with and without the *lox2/3* silencing construct (WT and *TPS10* v. *lox2/3* and *lox2/3xTPS10*, $W_{15,15}$ = 171.5, corrected p=0.035). No significant effect of population type on emission was observed (Wilcoxon rank sum tests at day or night of individuals vs center plants in monocultures, corrected p-values >0.07).

More than 18 times as much TAB and TBF were measured in *TPS10* and *lox2/3xTPS10* tissue extracts than in WT tissue extracts from field-grown plants. Significant differences (corrected p-values <0.04) in Wilcoxon rank sum tests between individual genotypes following Kruskal–Wallis tests (corrected p-values <0.001) are shown in **Figure 1—source data 3**. There was no significant effect of plant type or population type on extracted TAB and TBF (corrected p-values >0.8 in Kruskal–Wallis tests across individuals, edge and center plants, or across individuals, mono-, and mixed cultures). Emission of TAB and TBF from plants with natural background levels of herbivore attack in season one was also increased in *TPS10* and *lox2/3xTPS10*, to more than 20-fold WT levels. During the day, when emission of TAB and TBF was highest, Kruskal–Wallis tests indicated highly significant differences among genotypes (TAB, $\chi^2_3$ = 20.90, corrected p=0.008; TBF, $\chi^2_3$ = 19.63, corrected p=0.001); significant pairwise differences (corrected p<0.05) in Wilcoxon rank sum tests are shown in **Appendix 2—Table 1**. At night, when TAB and TBF emission levels were 10- to 20-fold lower overall, genotypes with and without the *TPS10* overexpression construct still differed significantly in TAB emission in a Wilcoxon rank sum test (WT and *lox2/3* v. *TPS10* and *lox2/3xTPS10*, $W_{15,15}$ = 52.5, corrected p=0.022). No significant effect of population type on emission was observed (Wilcoxon rank sum tests at day or night of individuals vs center plants in monocultures, corrected p-values >0.3).

**Appendix 2—Table 1.** Volatiles (percent IS plant$^{-1}$, mean ± SEM) trapped in the headspace around single plants in experimental season one (June 8th–9th).

| Genotype | Day n | Night n | GLVs (percent IS plant$^{-1}$) (Z)-Hexen-3-ol Day | Night | TPS10 products (percent IS plant$^{-1}$) TAB Day | Night | TBF Day | Night | Non-target volatiles (percent IS plant$^{-1}$) α-Duprezianene Day | Night | Germacrene A Day | Night |
|---|---|---|---|---|---|---|---|---|---|---|---|---|
| WT | 8 | 8 | 0.99% ± 0.99% | 7.96% ± 4.25% | 0.37% ± 0.29% a | — | — | — a | 2.13% ± 0.85% | 0.96% ± 0.42% | 1.68% ± 0.98% | — |
| TPS10 | 7 | 7 | 2.37% ± 1.55% | 2.84% ± 0.63% | 18.97% ± 6.03% b | 0.94% ± 0.53% | 9.34% ± 3.44% | 0.20% ± 0.20% b | 9.47% ± 5.26% | 0.94% ± 0.26% | 2.78% ± 1.52% | — |
| lox2/3 | 7 | 8 | 0.13% ± 0.13% | 1.06% ± 0.64% | — a | 0.15% ± 0.15% | — | — a | 2.75% ± 1.12% | 0.60% ± 0.19% | 1.68% ± 1.49% | — |
| lox2/3xTPS10 | 7 | 7 | 0.07% ± 0.07% | 1.24% ± 0.84% | 7.39% ± 2.56% b | 2.08% ± 0.84% | 4.47% ± 1.70% | 0.40% ± 0.40% b | 3.02% ± 1.42% | 0.73% ± 0.31% | 0.66% ± 0.37% | — |

(TAB Day column grouping annotations: $a_{lox}$, $b_{lox}$; TBF Day column grouping annotations: $a_{TPS}$, $b_{TPS}$)

a, b Different letters indicate significant differences between lines for a vertical category (corrected P<0.05) in Wilcoxon rank sum tests following significant Kruskal-Wallis tests across all genotypes (see Results text); P-values were corrected for multiple testing using the Holm-Bonferroni method. Community type was also tested and found not significant. Where there are no letters, there are no significant pairwise differences within a category.

$a_{lox}$ / $a_{TPS}$, $b_{lox}$, $b_{TPS}$ Indicate significant differences (corrected P<0.05) between lines with and without either the lox construct (WT and TPS v. lox and loxTPS) or the TPS construct (WT and lox v. TPS and loxTPS) in Wilcoxon rank sum tests for a category; P-values were corrected for multiple testing using the Holm-Bonferroni method. There are no significant pairwise differences in these categories.

IS, internal standard; GLVs, green leaf volatiles; TAB, (E)-α-bergamotene; TBF, (E)-β-farnesene; —, not detected.

In summary, all measures of GLV emission, jasmonate-mediated defense, and TAB and TBF emission revealed that the engineered phenotypes were stable and, except for open headspace measurements from *lox2/3* plants in mixed cultures, unaffected by population type in field experiments (*Figure 1C*, *Figure 1—source data 3*, *Appendix 2—Table 1*). Furthermore, non-target volatiles (phenylpropanoids, monoterpenes, and the sesquiterpenes germacrene A and α-duprezianene) were not affected by transformation or population type (*Figure 1—source data 3*, *Appendix 2—Table 1*). Thus, there was no evidence for direct effects of TPS10 volatiles on neighbor plants, with the exception of headspace contamination (*Figure 3*) which could be due either to diffusion or adsorption and re-release of TPS10 volatiles.

## Appendix 3

Screening of transgenic plant lines, quantification of TPI activity in field tissue samples, and quantification of volatiles in the headspace of field-grown plants (data shown in Appendices 1, 2); and statistical models used to test the hypotheses that *LOX2/3* deficiency, *TPS10* expression, plant position, and population type were associated with differences in foliar herbivore damage (data shown in **Figure 5**) and foliar herbivore abundance (data shown in **Figure 6**).

## Screening of transgenic plant lines

Transgenic lines were screened based on the procedure described by **Gase et al. (2011)**. Briefly, plants likely to contain a single transgene insertion were selected based on the segregation of hygromycin resistance in the T1 generation (the *HYGROMYCIN PHOSPHOTRANSFERASE II* [*HPTII*] gene from pCAMBIA-1301 is contained in the pSOL vector). High-quality genomic DNA was extracted using a modified cetyltrimethylammonium bromide (CTAB) method (**Bubner et al., 2006**) from non-senescent leaf tissue collected from transformed plants (see 'Tissue samples from glasshouse- and field-grown plants'). Single transgene insertions were confirmed by Southern blotting of genomic DNA digested with EcoRI and XbaI using a $^{32}P$-labeled probe binding to *HPTII* (*lox2/3* shown in **Appendix 1— Figure 1**, *TPS10* in **Schuman et al., 2014**). Flow cytometric analysis (described by **Bubner et al., 2006**) confirmed that all transgenic lines were diploid. Homozygosity of T2 plants was determined by screening for resistance to hygromycin.

The accumulation or reduction of target gene transcripts was quantified in leaf samples (n = 4) harvested 1 h after treatment with *M. sexta* oral secretions (W + OS, see 'Tissue samples from glasshouse- and field-grown plants' in 'Materials and methods'). Total RNA was extracted from 100 mg tissue using TRI reagent (Invitrogen, Life Technologies, Carlsbad, CA) according to the manufacturer's instructions. Quality was checked on a 1% agarose gel and concentration was determined by absorbance at 260 nm with a NanoDrop ND-1000 spectrophotometer (NanoDrop Technologies, Wilmington, DE). Synthesis of cDNA from 0.5 µg of total RNA per sample and SYBR Green qPCR analyses were conducted as in Wu et al. (2007) using reagents from Fermentas, Waltham, MA, a Mastercycler (Eppendorf, Germany) and an Mx3005P qPCR system (Stratagene La Jolla, CA) and qPCR Core Kit for SYBR Green I (Eurogentec, Belgium). Transcripts were quantified using external standard curves for each gene, and *N. attenuata ELONGATION FACTOR 1A* (*Elf1A*) transcript abundance in each sample was used to normalize total cDNA concentration variations. Samples of RNA used to make cDNA were pooled to the same dilution as in cDNA samples and run alongside cDNA in all qPCRs to control for gDNA contamination; no contamination was detected. The primers for each gene have been described previously: *LOX2*, **Allmann et al. (2010)**; *LOX3*, **Kallenbach et al. (2010)**; and *TPS10* and *Ef1A*, **Schuman et al. (2014)**.

Analysis of JA and JA-Ile is described under 'Quantification of JA and JA-Ile in leaf tissue' in 'Materials and methods'.

## Quantification of TPI activity in field tissue samples

TPI activity (nmol mg protein$^{-1}$) was quantified in 100 mg of frozen tissue from field samples (n = 11–24) using a radial diffusion assay as previously described (*van Dam et al., 2001*) based on the protocol from *Jongsma et al. (1993)* and using materials from Sigma–Aldrich and a radial plate sufficiently large to simultaneously accommodate all samples and two rows of standards, one at the start and one at the end of the plate. For the statistical analysis approach, see 'Statistical analyses' in 'Materials and methods'.

## Quantification of volatiles in the headspace of field-grown plants

For volatile trapping in field experiments, we used single-use activated charcoal filters (ORBO-32 standard; Supelco, Sigma-Aldrich) shielded with aluminum foil from UV radiation and connected to self-made ozone scrubbers at the air inlet that contained eight-ply of 65 mm-diameter MnO2-coated copper gauze (OBE Corporation, Fredericksburg, TX) to prevent the oxidation of volatiles by ozone (*Calogirou et al., 1996*). Ambient air was pulled through the charcoal filter by a vacuum pump (DAA-V114-GB; Gast, Benton Harbor, MI) powered by a car battery. After trapping, charcoal filters were sealed with provided caps and stored at −20°C until transportation to the MPICE, elution, and analysis. Filters were spiked with tetralin internal standard (IS) and eluted as described under 'Quantification of volatiles in the headspace of glasshouse-grown plants'.

For the volatile data in *Appendix 2—Table 1*, heat- and light-resistant transparent plastic cones with open tops (frost protection cones, Merox) were placed over individual plants and plants at the centers of monocultures (n = 7–8 plants per genotype) on June 8th in season one, and the charcoal filter was placed at the base of the plant such that air flowed in at the top of the cone, through the headspace, and out through the filter. Airflow was started at 8:45 am. Volatiles were trapped for a total of 22 h and filters were exchanged after 12 h, resulting in two samples from each plant for the periods from 9 am to 9 pm (day) and 9 pm to 7 am (stopped before daylight shone on plants, night). Samples were analyzed by a Varian *C*P-3800 GC-Saturn 4000 ion trap MS connected to a ZB5 column (30 m × 0.25 mm i.d., 0.25 μm film thickness; Phenomenex) as described in 'Quantification of volatiles in the headspace of glasshouse-grown plants'. Analyte quantities were expressed as percent tetralin IS per plant. For the statistical analysis approach, see 'Statistical analyses' in 'Materials and methods'.

## Statistical models

### Statistical models used for data in *Figure 5*

We tested the hypotheses that *LOX2/3* deficiency, *TPS10* expression, plant position, and population type were associated with differences in foliar herbivore damage. ANOVAs or linear mixed-effects models were performed on arcsine-transformed data. Models were fit by maximum likelihood and model reduction was performed by removing least significant terms and comparing the Akaike information criterion (AIC) value of more complex vs simplified models (*Akaike, 1973*) until only significant terms were left. Linear mixed-effects models were re-fit by restricted maximum likelihood (REML). Genotypes were coded as follows: 1WT, 2TPS10, 3lox23, 4lox23xTPS10; *LOX2/3* deficiency was coded as 1WT, 2 lox; *TPS10* expression was coded as 1WT, 2TPS; plant positions were coded as 1Individual, 2Edge, 3Center; and monocultures vs mixed cultures were coded as 1Mono, 2Mix. All minimal models to explain differences in herbivore damage retained *LOX2/3* deficiency as the only significant factor, although there was a marginal interaction between *LOX2/3* deficiency and plant position in season one which resulted in a poorer AIC value when removed.

For individual plants, the following initial ANOVA model was used:

model1<-aov(Arcsin_total ~ lox*TPS,ind),

and the final model was

model3<-aov(Arcsin_total ~ lox,ind)

with the following results:

|  | Df | Sum sq | Mean sq | F | p |
|---|---|---|---|---|---|
| **Season 1** | | | | | |
| *LOX2/3* deficiency | 1 | 0.1008 | 0.1008 | 11.47 | **0.0015** |
| Residuals | 44 | 0.3864 | 0.0088 | | |
| **Season 2** | | | | | |
| *LOX2/3* deficiency | 1 | 0.2321 | 0.2321 | 14.64 | **0.0005** |
| Residuals | 39 | 0.6183 | 0.0159 | | |

For edge plants, a linear mixed-effects model was used to test the effects of *LOX2/3* deficiency and *TPS10* expression, so that the analysis of multiple plants per population could be included as a random blocking factor (lme function in R package nlme, **Pinheiro et al., 2014**). The initial model was

model1<-lme(Arcsin_total ~ lox*TPS,random= ~ 1|PopID,edge,method="ML")

and the final model was

model3REML<-lme(Arcsin_total ~ lox,random= ~ 1|PopID,edge)

with the following results (fixed effects):

|  | Value | Std. error | DF | t | p | (Intr) |
|---|---|---|---|---|---|---|
| **Season 1** | | | | | | |
| (Intercept) | 0.1929 | 0.0102 | 200 | 18.86 | **<0.0001** | |
| *LOX2/3* deficiency | 0.0895 | 0.0145 | 70 | 6.152 | **<0.0001** | −0.704 |
| **Season 2** | | | | | | |
| (Intercept) | 0.31 | 0.02 | 119 | 20.40 | **<0.0001** | |
| *LOX2/3* deficiency | 0.13 | 0.02 | 74 | 5.501 | **<0.0001** | −0.641 |

Furthermore, WT and *lox2/3* edge plants were tested to determine whether levels of herbivore damage differed with the presence or absence of a *TPS10*-expressing plant in the population, starting with the following model:

model1<-lme(Arcsin_total ~ lox*MonoMix,random= ~ 1|PopID,edgeWTlox,method="ML")

and the final model was

model6REML<-lme(Arcsin_total ~ lox,random= ~ 1|PopID,edgeWTlox),

with the following results (fixed effects):

|  | Value | Std. error | DF | t | p | (Intr) |
|---|---|---|---|---|---|---|
| **Season 1** | | | | | | |
| (Intercept) | 0.1947 | 0.0125 | 130 | 15.60 | **<0.0001** | |
| *LOX2/3* deficiency | 0.0945 | 0.0178 | 46 | 5.309 | **<0.0001** | −0.702 |
| **Season 2** | | | | | | |
| (Intercept) | 0.3132 | 0.0194 | 71 | 16.16 | **<0.0001** | |
| *LOX2/3* deficiency | 0.1454 | 0.0307 | 46 | 4.732 | **<0.0001** | −0.631 |

For center plants, the following initial ANOVA model was used to test the effects of *LOX2/3* deficiency and *TPS10* expression:

model1<-aov(Arcsin_total ~ lox*TPS,center)

and the final model was

model3<-aov(Arcsin_total ~ lox,center)

with the following results:

|  | Df | Sum sq | Mean sq | F | p |
|---|---|---|---|---|---|
| **Season 1** |  |  |  |  |  |
| *LOX2/3* deficiency | 1 | 0.1632 | 0.16325 | 23.57 | 8.28E-06 |
| Residuals | 63 | 0.4363 | 0.00692 |  |  |
| **Season 2** |  |  |  |  |  |
| *LOX2/3* deficiency | 1 | 0.2129 | 0.2129 | 6.545 | 0.0131 |
| Residuals | 60 | 1.952 | 0.0325 |  |  |

Furthermore, *TPS10* and *lox2/3xTPS10* center plants were tested to determine whether levels of herbivore damage differed in mixed- vs monocultures, with the initial model

model1<-aov(Arcsin_total ~ lox*MonoMix,centerTPSloxTPS)

and the final model

model3<-aov(Arcsin_total ~ lox,centerTPSloxTPS)

with the following results:

|  | Df | Sum sq | Mean sq | F | p |
|---|---|---|---|---|---|
| **Season 1** |  |  |  |  |  |
| *LOX2/3* deficiency | 1 | 0.0663 | 0.0663 | 8.726 | 0.0052 |
| Residuals | 40 | 0.3038 | 0.0076 |  |  |
| **Season 2** |  |  |  |  |  |
| *LOX2/3* deficiency | 1 | 0.1679 | 0.1679 | 4.47 | 0.0403 |
| Residuals | 43 | 1.6147 | 0.0376 |  |  |

Finally, to determine the effect of plant position, an ANOVA was conducted on individuals and edge and center plants from monocultures; values used for edge plants were the arithmetic mean of all edge plants in a population to reduce pseudoreplication, but a linear mixed-effects model was still used to correct for the fact that center and edge plants were from the same populations. The initial model was

model1<-lme(Mean_arcsin_total ~ lox*Position*TPS,random= ~ 1|PopID,ind_mono, method="ML")

and the final model in season one was

model6REML<-lme(Mean_arcsin_total ~ lox+Position,random= ~ 1|PopID,ind_mono)

with the following results:

LOX2/3 deficiency

| Season 1 | Value | Std. error | DF | t | p | (Intr) | X2lx2l | Position2Edge |
|---|---|---|---|---|---|---|---|---|
| (Intercept) | 0.19326 | 0.01414 | 92 | 13.66 | <0.0001 |  |  |  |

*continued on next page*

*continued*

| Season 1 | Value | Std. error | DF | t | p | (Intr) | X2lx2l | Position2Edge |
|---|---|---|---|---|---|---|---|---|
| *LOX2/3* deficiency | 0.09517 | 0.01592 | 92 | 5.978 | <0.0001 | −0.538 | | |
| Position2Edge | −0.00666 | 0.01668 | 42 | −0.3990 | 0.6919 | −0.591 | −0.021 | |
| Position3Center | −0.03057 | 0.01689 | 42 | −1.8097 | 0.0775 | −0.588 | −0.013 | 0.819 |

The final model in season two was

model7REML<-lme(Mean_Arcsin_Total ~ *LOX2/3* expression,random= ~ 1|PopID,ind_mono)

| Season 2 | Value | Std. error | DF | t | p | (Intr) |
|---|---|---|---|---|---|---|
| (Intercept) | 0.3222 | 0.0153 | 121 | 21.12 | <0.0001 | |
| *LOX2/3* deficiency | 0.1274 | 0.0227 | 121 | 5.619 | <0.0001 | −0.673 |

Statistical models used for data in *Figure 6*.

We tested the hypotheses that *LOX2/3* and *TPS* expression, plant position, and population type were associated with differences in foliar herbivore abundance. Generalized linear mixed-effects models were used to analyze data presented in *Figure 6A,B* so that the analysis of multiple plants per population could be included as a random blocking factor (glmer function in R in package lme4, *Bates et al., 2014a*; *Bates et al., in review*), while a generalized linear model (glm) was used for data in *Figure 6C* (in which case only one plant per population was analyzed). Models were fit by maximum likelihood (Laplace Approximation) using a poisson error distribution. Genotypes were coded as follows: 1WT, 2TPS10, 3lox23, 4lox23xTPS10; plant positions were coded as 1Individual, 2Edge, 3Center; and monocultures vs mixed cultures were coded as 1Mono, 2Mix.

For data shown in *Figure 6A*, the following model was used:

model<-glmer(Herbivores ~ Genotype+(1|PopID),counts_all,family=poisson)

with the following results:

| Fixed effects | Estimate | Std. error | z | p |
|---|---|---|---|---|
| (Intercept) | 1.7 | 0.2361 | 7.219 | 5.25E-13 |
| Genotype*TPS10* | −0.8589 | 0.2116 | −0.059 | 9.30E-05 |
| Genotype*lox2/3* | −0.612 | 0.3097 | −1.983 | 0.07 |
| Genotype*lox2/3xTPS10* | −1.877 | 0.097 | −0.582 | 6.00E-06 |
| **Correlation of fixed effects** | **(Intr)** | **GenotypeTPS10** | **Genotypelox2/3** | |
| Genotype*TPS10* | −0.36 | | | |
| Genotype*lox2/3* | −0.369 | 0.27 | | |
| Genotype*lox2/3xTPS10* | −0.325 | 0.22 | 0.531 | |

Tukey post-hoc contrasts were calculated using the multcomp package in R (*Hothorn et al., 2008*) as

modelposthoc <- glht(model1, linfct = mcp(Genotype = "Tukey"))

with the following results:

| Linear hypotheses | Estimate | Std. error | z | p |
|---|---|---|---|---|
| *TPS10*–WT = 0 | −0.8589 | 0.2116 | −4.059 | 0.0003 |
| *Lox2/3*–WT = 0 | −0.6142 | 0.3097 | −1.983 | 0.1839 |
| *Lox2/3 xTPS10*–WT = 0 | −1.877 | 0.4097 | −4.582 | <1e-04 |

*continued*

| Linear hypotheses | Estimate | Std. error | z | p |
|---|---|---|---|---|
| *lox2/3–TPS10 = 0* | 0.2447 | 0.3236 | 0.756 | 0.8675 |
| *lox2/3 xTPS10–TPS10 = 0* | −1.019 | 0.4168 | −2.444 | 0.0644 |
| *lox2/3 xTPS10–LOX2/3 = 0* | −1.263 | 0.3592 | −3.517 | **0.0023** |

For data shown in *Figure 6B*, the following model was used:

model<-glmer(Herbivores ~ Genotype*Position+(1|PopID), counts_WT_*TPS10*_Mono_Ind, family=poisson)

with the following results:

| Fixed effects | Estimate | Std. error | z | p |
|---|---|---|---|---|
| (Intercept) | 1.961 | 1.063 | 1.844 | **0.0652** |
| Genotype*TPS10* | −1.122 | 0.3064 | −3.662 | **0.0003** |
| PositionEdge | −0.8305 | 1.166 | −0.712 | 0.4762 |
| PositionCenter | −1.363 | 1.193 | −1.143 | 0.2532 |
| Genotype*TPS10*:PositionEdge | 0.2737 | 0.7789 | 0.351 | 0.7253 |
| Genotype*TPS10*:PositionCenter | 1.748 | 0.7995 | 2.186 | **0.0288** |

| Correlation of fixed effects | (Intr) | GenotypTPS10 | PositionEdge | PositionCenter | G*TPS10*:PE |
|---|---|---|---|---|---|
| Genotyp*TPS10* | −0.071 | | | | |
| PositionEdge | −0.912 | 0.065 | | | |
| PositionCenter | −0.891 | 0.063 | 0.955 | | |
| G*TPS10*:PE | 0.028 | −0.393 | −0.269 | −0.228 | |
| G*TPS10*:PC | 0.027 | −0.383 | −0.231 | −0.323 | 0.797 |

For data shown in 6C, the following initial model was used:

model1<-glm(Herbivores ~ Genotype*MixMono,counts_WT_lox23_edge,family=poisson),

and the model was reduced to

model3<-glm(Herbivores ~ Genotype,counts_WT_lox23_edge,family=poisson),

wit the following results:

| Fixed effects | Estimate | Std. error | z | p |
|---|---|---|---|---|
| (Intercept) | 1.421 | 0.1313 | 10.83 | **<2e-16** |
| Genotype*lox2/3* | 0.4504 | 0.191 | 2.358 | **0.0183** |

