## [Decision Letter]

Thank you for sending your work entitled “Plant defense phenotypes determine the consequences of volatile emission for individuals and neighbors” for consideration at *eLife*. Your article has been favorably evaluated by Diethard Tautz (Senior editor), a Reviewing editor, and three reviewers.

The Reviewing editor and the reviewers discussed their comments before we reached this decision, and the Reviewing editor has assembled the following comments to help you prepare a revised submission.

This manuscript makes an important contribution to understanding the role of two volatiles. The reviewers identified a few issues that need improvement.

1) You report that the role of *lox* mediated direct defence and its associated volatiles is determining damage on the plot level, not affecting focal or neighbouring plants. Whereas the *TSP10* plants do affect abundance of stem borer beetles and predator presence on individual plant level. This conclusion is sound based on the data presented, but it is also biased by the experimental design. You did not explore the role of *LOX* derived volatiles or direct defences on the individual plant level experimentally. You should acknowledge and discuss the fact that the effect of *LOX* mediated effects has not been tested fully. The Discussion should include how the missing treatment might affect your conclusions or further inferences about the role of particular volatile emissions.

2) Although from the plants perspective measurement of damage is a useful predictor of the costs of herbivory, the measurement may not accurately present the actual effect on herbivore abundance. Damage is confounding: herbivore abundance, their feeding intensity (that individuals may adjust to plant quality), and predator pressure on herbivores. Actual herbivore counts such as provided for the stem borer beetles would more strongly support the conclusions on *LOX* and *TSP* mediated community processes. Please discuss this point or provide the counts if you have them.

3) You rely very heavily on pairwise comparisons between treatments, usually with non-parametric tests or G tests for differences in proportions. The results would be much clearer to the readers, and avoid many of the concerns over multiple comparisons, if the authors used generalized linear models that better reflect the experimental design (see below for specific suggestion). For instance, the strongest results for their “community genetics” argument comes from the pattern of reduced mortality of *lox2/3* vs. WT plants when in the presence of a *TPS10* plant (Figure 9), but you do not test this directly. This pattern should be tested as an interaction between the *lox* and *TPS* treatments. You interpret this pattern strongly, but all of the tests performed show marginal or non-significant differences (a marginal decline in mortality for *lox2/3* plants and a non-significant increase in mortality for WT plants due to the presence of a *TPS10* plant). A similar model could be used for stem borer infestation and predation rates. In addition to providing more powerful tests, it may help the reader if all of these analyses are presented in a parallel way.

Specific suggestion: your statistical analysis could be improved for the analysis of the field experiment data (mortality, prevalence of stem borers, and prevalence of predation).

All of the outcome variables are binomial (1 or 0 at the level of an individual plant). It is recommended that you model this outcome with a generalized linear mixed model, using a binomial error distribution. This could be done with the glmer function in R or the GLIMMIX procedure in SAS, among other options. The model would include two crossed factors: the *LOX* genotype of the individual plant (2 levels, wild type or mutant) crossed with the *TPS10* treatment of the population (5 levels, individual WT, individual *TPS10*, monoculture WT, monoculuture *TPS10*, mixed), and plot as a random effect to account for the non-independence of plants from the same plot. The test of whether the presence of a *TPS10* plant alters mortality on non-*TPS10* neighbors differently based on the *LOX* genotype of those neighbors would come from the interaction term in this model, with the *TPS10* producing plant individuals excluded (and thus the *TPS10* monoculture treatment dropped). Or alternatively, you could drop the individual plant treatments and have a three way factorial model of *LOX* genotype*population treatment (mono WT, mono *TPS10*, mixed)*position (Center vs. Edge). You could then perform an a priori contrast to compare the interaction specifically between the (WT monoculture vs. mixture)*(*LOX* genotype) for edge plants to test the significance of the pattern in Figure 9.

4) Please consider that a streamlining of the presentation would make the manuscript easier to read and understand. A great deal of space is devoted to establishing that the gene silencing/overexpression worked and produced reliable phenotypes in controlled conditions. These methods and results could be moved to an appendix. This would allow the authors to focus on the results from the field experiment, where the interesting finding is not just that manipulating plant defenses and volatiles affects herbivore damage and plant fitness, but that the genotype of one individual can affect the outcome on neighboring individuals.

[Editors' note: further revisions were requested prior to acceptance, as described below.]

Thank you for resubmitting your work entitled “Plant defense phenotypes determine the consequences of volatile emission for individuals and neighbors” for further consideration at *eLife*. Your revised article has been favorably evaluated by Diethard Tautz (Senior editor), a member of the Board of Reviewing Editors, and two of the original reviewers. The manuscript has been improved but there are some remaining issues that need to be addressed before acceptance, as outlined below.

The reviewers found your manuscript to be improved. However there remains a statistical issue that requires resolution. This was framed by one of the reviewers as follows:

”My other concern with the original article was that the authors used a great deal of pairwise comparisons rather than building statistical models that better reflected their experimental design and could test directly for the interactive effects at the heart of their arguments. The authors made some attempt to respond to this suggestion, providing revamped analyses for plant size, flower number, and herbivore damage. However, I was disappointed to see that they did not collect their data in a way that would allow this more powerful analyses of the plant mortality data (which is the clearest case of “extended phenotypes” where the genotype of the center plant affected the fitness of the edge plants), or the data on stem borer abundance and predation rates. For the mortality data especially, this is surprising to me: why would the authors not have noted the live/dead status of plants individually, rather than just taking a tally? Do they mean that within a given population of five plants, they tallied the total number dead (1-5), or do they really mean that they cannot ascribe their mortality data even to the plot level (the base unit of replication for their experiment). If it is the former, they could still use a binomial generalized linear model, as long as they kept the data from the center and edge plants separately. If it is the latter, this is concerning to me because this analysis does not reflect the psuedoreplication induced by having five plants per plot. While a G test is “unreplicated” in that you need only enter a single frequency, it still assumes that the individuals comprising those frequencies are independent observations (i.e. a comparison between 28/75 vs. 50/75 is more powerful than 6/15 vs. 10/15). And I am truly puzzled as to how the authors can have data on flower production at an individual plant scale, but not have mortality data on that scale. Surely if they know how many flowers any given plant produced, they must know if it was alive or not. Am I missing something obvious here?”

In light of these comments, please clarify how the data on plant mortality and stem borer infection was collected. If the mortality analysis is unreplicated as described in the cover letter, please justify why it is still appropriate to use the statistical test that you used.

---

## [Author Response]

*1) You report that the role of lox mediated direct defence and its associated volatiles is determining damage on the plot level, not affecting focal or neighbouring plants. Whereas the* TSP10 *plants do affect abundance of stem borer beetles and predator presence on individual plant level. This conclusion is sound based on the data presented, but it is also biased by the experimental design. You did not explore the role of* LOX *derived volatiles or direct defences on the individual plant level experimentally. You should acknowledge and discuss the fact that the effect of* LOX *mediated effects has not been tested fully. The Discussion should include how the missing treatment might affect your conclusions or further inferences about the role of particular volatile emissions*.

We agree that we did not always explicitly consider the limits of our experimental design when discussing the data, and to correct this, we have added the following text to the Discussion:

a) “It should be noted that our experimental design did not support the analysis of population-level effects […] in the presence, or absence, of total HIPVs and jasmonate-mediated defense.”

b) We changed the subsection title from “*LOX2/3* expression has large direct effects on plants, while the largest effects of *TPS10* expression are indirect” to “*LOX2/3* expression has large direct effects on plants, while *TPS10* expression has large indirect effects”.

c) We have also added the following sentence: “In contrast, due to the large suite of defenses regulated by *LOX2* and *LOX3* products, it would have been more difficult to determine the proximate cause of neighbor effects resulting from manipulation of *LOX2* and *LOX3* expression.”

d) We had already stated in the Discussion: “Although several components of plant HIPV blends have been shown to prime the direct and indirect defenses of neighbors in different systems (reviewed in [8]; [7]; and see Heil, Kost 2006), GLVs may often be the active cues (27; 51; 29). We did not mix GLV-deficient with GLV-emitting plants in our experimental design, but we can conclude that *TPS10* volatiles had no detectable direct effects on neighbor plants in our set-up.” (This was in the first submission and has not changed.)

e) “Of course it is possible that the manipulation of *LOX2* and *LOX3* expression within populations would also reveal such extortionist-like effects, for example if *T. mucorea* preferred to oviposit in *TPS10* plants over *LOX2/3*x*TPS10* plants in the same population, which would support the hypothesis that *T.* mucorea preference reflects the ratio between *TPS10* volatiles and other HIPVs.”

f) And finally, we added: “In this context, it would be interesting to know whether central *LOX2/3* plants surrounded by WT edge plants would experience a similar predation disadvantage as central *TPS10* plants surrounded by WT edge plants, and whether central WT plants surrounded by *LOX2/3* edge plants would have an advantage over their neighbors in attracting predators.”

*2) Although from the plants perspective measurement of damage is a useful predictor of the costs of herbivory, the measurement may not accurately present the actual effect on herbivore abundance. Damage is confounding: herbivore abundance, their feeding intensity (that individuals may adjust to plant quality), and predator pressure on herbivores. Actual herbivore counts such as provided for the stem borer beetles would more strongly support the conclusions on* LOX *and* TSP *mediated community processes. Please discuss this point or provide the counts if you have them*.

We thank the reviewers for this astute suggestion. We do indeed have one time point of herbivore count data and had previously not included it because the data are from only a subset of plants and we considered that they might not be as robust as the damage data. However, we re-visited these data and have now included them as a new Figure 6 (the original figures have been re-numbered accordingly). Indeed, the herbivore abundance data differs from the damage data. This must be interpreted with caution as the abundance data were recorded on June 26^th^ while the damage data were recorded on June 9^th^, but we state these dates clearly in the text and note, in the Results and in the Discussion, that observation of the plants indicated that there were no changes in trends of herbivore damage between these dates (i.e., relative damage levels remained consistent throughout the end of the experiment). These data are also introduced in the Introduction, reviewed in the Results and in the Discussion, in several commented locations, and a new paragraph has been included in the Methods under the subsection headed “Herbivore and predator counts”.

*3) You rely very heavily on pairwise comparisons between treatments, usually with non-parametric tests or G tests for differences in proportions. The results would be much clearer to the readers, and avoid many of the concerns over multiple comparisons, if the authors used generalized linear models that better reflect the experimental design (see below for specific suggestion). For instance, the strongest results for their “community genetics” argument comes from the pattern of reduced mortality of* lox2/3 *vs. WT plants when in the presence of a* TPS10 *plant (*Figure 9*), but you do not test this directly. This pattern should be tested as an interaction between the lox and* TPS *treatments. You interpret this pattern strongly, but all of the tests performed show marginal or non-significant differences (a marginal decline in mortality for* lox2/3 *plants and a non-significant increase in mortality for WT plants due to the presence of a* TPS10 *plant). A similar model could be used for stem borer infestation and predation rates. In addition to providing more powerful tests, it may help the reader if all of these analyses are presented in a parallel way*.

Specific suggestion.: your statistical analysis could be improved for the analysis of the field experiment data (mortality, prevalence of stem borers, and prevalence of predation).

*All of the outcome variables are binomial (1 or 0 at the level of an individual plant). It is recommended that you model this outcome with a generalized linear mixed model, using a binomial error distribution. This could be done with the glmer function in R or the GLIMMIX procedure in SAS, among other options. The model would include two crossed factors: the* LOX *genotype of the individual plant (2 levels, wild type or mutant) crossed with the* TPS10 *treatment of the population (5 levels, individual WT, individual* TPS10*, monoculture WT, monoculuture* TPS10*, mixed), and plot as a random effect to account for the non-independence of plants from the same plot. The test of whether the presence of a* TPS10 *plant alters mortality on non-*TPS10 *neighbors differently based on the LOX genotype of those neighbors would come from the interaction term in this model, with the* TPS10 *producing plant individuals excluded (and thus the* TPS10 *monoculture treatment dropped). Or alternatively, you could drop the individual plant treatments and have a three way factorial model of* LOX *genotype*population treatment (mono WT, mono* TPS10*, mixed)*position (Center vs. Edge). You could then perform an a priori contrast to compare the interaction specifically between the (WT monoculture vs. mixture)*(*LOX *genotype) for edge plants to test the significance of the pattern in*Figure 9.

We have revisited these raw data and regrettably, none of these data sets were collected in a manner which we think permits application of the modeling techniques suggested. Specifically, the *Trichobaris mucorea* infestation data and mortality data were simply collected as tallies per individual or population type and genotype at the end of the season, i.e., not replicated. Thus an unreplicated analysis such as the G-test, according to our understanding, must be used. The predation assays were not always conducted on exactly the same group of plants, as over the course of the 5 repetitions some plants were damaged by wind or severed by cutworms and new plants had to be assigned. Thus it would be difficult to appropriately apportion error to a random or repeated-measures term in a model, and we feel it is most appropriate for these data as well to treat them as total frequencies and conduct an unreplicated statistical analysis. We have stated these limitations clearly in the Methods and Results sections describing the relevant experiments, as marked by comments in the revised text.

However, we greatly appreciate the reviewers’ points and detailed suggestions. We thus reviewed all of the data in the manuscript and determined that other data sets would be more appropriately analyzed with such an approach. We therefore used an approach similar to that suggested by the reviewers to analyze the newly included foliar herbivore abundance data, and we furthermore re-analyzed our foliar herbivore damage, plant size, and flower production data. These changed analyses are presented in the new version of Figure 5 (foliar herbivore damage data), the new Figure 6 and source data files (foliar herbivore abundance data), and a new Figure 8 and source data file (re-analyzed size and flower production data). The general statistical approaches are described in the revised statistical analysis section of the Materials and methods, and in addition, we have provided detailed descriptions of the analyses as part of the new Appendix 3.

In particular, the re-analysis of plant size and reproduction data allowed us to simplify the presentation of the flowering data in Figure 7 to focus only on *LOX2/3* deficiency effects, while revealing subtler interactive effects of *TPS10* expression in the separate effects (new analysis presented in the new Figure 8 and [Supplementary-material SD8-data]), which fit well with the other data sets, and which were overlooked in our previous analysis.

Overall, the revised statistical analyses have revealed more population-level and interactive effects of *TPS10* expression which we feel facilitate our discussion of the data, strengthen our conclusions, and overall provide a more comprehensible and coherent picture for the reader.

*4) Please consider that a streamlining of the presentation would make the manuscript easier to read and understand. A great deal of space is devoted to establishing that the gene silencing/overexpression worked and produced reliable phenotypes in controlled conditions. These methods and results could be moved to an appendix. This would allow the authors to focus on the results from the field experiment, where the interesting finding is not just that manipulating plant defenses and volatiles affects herbivore damage and plant fitness, but that the genotype of one individual can affect the outcome on neighboring individuals*.

We thank the reviewers for this suggestion to streamline the text. The *eLife* staff suggested that we create three appendices, each with one coherent topic or set of contents. Thus, the results regarding screening of transgenic lines are now in Appendix 1, results demonstrating field-stable phenotypes are in Appendix 2, and the corresponding methods are in Appendix 3. We feel this has greatly improved the readability and organization of the manuscript.

[Editors' note: further revisions were requested prior to acceptance, as described below.]

The reviewers found your manuscript to be improved. However there remains a statistical issue that requires resolution. This was framed by one of the reviewers as follows:

“My other concern with the original article was that the authors used a great deal of pairwise comparisons rather than building statistical models that better reflected their experimental design and could test directly for the interactive effects at the heart of their arguments. The authors made some attempt to respond to this suggestion, providing revamped analyses for plant size, flower number, and herbivore damage. However, I was disappointed to see that they did not collect their data in a way that would allow this more powerful analyses of the plant mortality data (which is the clearest case of “extended phenotypes” where the genotype of the center plant affected the fitness of the edge plants), or the data on stem borer abundance and predation rates. For the mortality data especially, this is surprising to me: why would the authors not have noted the live/dead status of plants individually, rather than just taking a tally? Do they mean that within a given population of five plants, they tallied the total number dead (1-5), or do they really mean that they cannot ascribe their mortality data even to the plot level (the base unit of replication for their experiment). If it is the former, they could still use a binomial generalized linear model, as long as they kept the data from the center and edge plants separately. If it is the latter, this is concerning to me because this analysis does not reflect the psuedoreplication induced by having five plants per plot. While a G test is “unreplicated” in that you need only enter a single frequency, it still assumes that the individuals comprising those frequencies are independent observations (i.e. a comparison between 28/75 vs. 50/75 is more powerful than 6/15 vs. 10/15). And I am truly puzzled as to how the authors can have data on flower production at an individual plant scale, but not have mortality data on that scale. Surely if they know how many flowers any given plant produced, they must know if it was alive or not. Am I missing something obvious here?”

*In light of these comments, please clarify how the data on plant mortality and stem borer infection was collected. If the mortality analysis is unreplicated as described in the cover letter, please justify why it is still appropriate to use the statistical test that you used*.

This is an important concern and a considered comment, and we want to give a correspondingly careful and reasoned response, which is below. In summary, the data presented in Figure 9 is the most biologically meaningful plant mortality data, which is unfortunately from an unreplicated data set; however the analysis presented in Figure 9 is appropriate for these data and our question, and does not violate the assumptions of the G-test regarding independence between the groups compared. Likewise, the assumption of independence is not violated for the analysis in Figure 11. However the assumption of independence of the Fisher’s test was violated in one instance for the analysis presented in Figure 10. We have corrected our analysis for Figure 10 and associated parts of the Results text, but the corrected analysis produces exactly the same conclusions.

Question 1: Should we replace our unreplicated plant mortality data set (presented in Figure 9) with a replicated data set, collected concurrently with flowering data?

Of course we did note plant mortality while collecting the data on flowering. However the flowering data was collected on May 29^th^ and June 6^th^, while we wished to analyze mortality by the end of the season on June 29^th^. All other data sets collected between June 6^th^ and June 29^th^ (herbivore abundance, predation rates) were collected on subsets of plants and plant mortality was not noted again until June 29^th^, at which point all surviving plants were tallied in categories while checking stems for infestation by *Trichobaris mucorea*. Thus total mortality for the season was calculated on June 29^th^ as the number of plants originally planted minus the number surviving in each category. Because these total mortality numbers from June 29^th^ most accurately reflect the likelihood that plants would not have survived to produce ripe seed, we feel it is most meaningful to present the June 29^th^ data.

One could argue that, although the final mortality numbers from June 29^th^ are principally more meaningful, nevertheless if mortality did not change much between June 6^th^ and June 29^th^, it would be preferable to present a replicated rather than an unreplicated data set. However there was a 1.5-fold increase in overall mortality, from 29% to 46%, between June 6^th^ and June 29^th^. Although the mortality data from June 6^th^ show similar trends as June 29^th^, with *LOX2/3* deficiency tending to increase mortality and *TPS10* expression tending to lower mortality, the differences on June 6^th^ are not significant even when analyzed using a GLM with a binomial distribution, because the total mortality is lower. Thus the June 6^th^ dataset has lower statistical power and, more importantly, is truly less biologically meaningful.

Answer to question 1: No. It is best to present the unreplicated June 29^th^ data.

Question 2: Is the use of G-tests to analyze the data in Figure 9 appropriate? What about the use of G-tests and Fisher’s tests in other data sets (Figure 10 and Figure 11)?

In analyzing total mortality on June 29^th^, we wished to know whether mortality rate was dependent on, or independent of, population type. We thus used G-tests of independence and conducted post-hoc P-value corrections for multiple comparisons as necessary, which we understand to be the most appropriate choice of tests for our unreplicated data and our question. Although our data came from experimental populations (plots), we did not violate the assumption of independence incorporated into the G-test for the data shown in Figure 9. This is because, in Figure 9, we only compare data from separate population types (separate plots), and we never compare data from plants at different positions in the same population type. If we had compared e.g. center to edge plants, there would be dependencies within the data set because some of the plants in group “edge” would have come from the same populations as plants in group “center”, and thus we would not be comparing independent observations. However by comparing different population types, we ensured that no plants in group A shared a population with any plants in group B, and therefore there are no dependencies between groups: they are independent. The same reasoning holds for the comparisons in Figure 11, and for the comparisons of mortality between plants differing only in *LOX2/3* deficiency or from different population types made in the Results section text (in the subsection headed “*LOX2/3* deficiency increases plant mortality, but mortality is mitigated by single *TPS10*-expressing plants in *LOX2/3-*deficient populations”).

For Figure 10, we compare both plants from separate population types and plants from the same population type. The comparison of plants from the same population type violates the independence assumption of the Fisher’s test and we thank the Reviewer for bringing this to our attention. We have removed the incorrect comparisons, changed the representation of the statistical comparisons in Figure 10 accordingly, and corrected the relevant parts of the figure caption and of the manuscript (subsection headed “*TPS10*-expressing plants increase the infestation rates of their neighbors by the stem-boring weevil *Trichobaris mucorea*”). In this part of the manuscript text, we also removed the Fisher’s tests of infestation by genotype: when comparing genotypes differing only in *TPS10* expression overall, rather than different population types, we also violated the assumption of independence because some populations contain both *TPS10*-expressing and non-expressing individuals. As a result of removing some of the pairwise comparisons, our post-hoc correction factor decreased and the *P*-value we report for the comparison of WT plants in mixed versus monocultures decreased from 0.025 to 0.015. Because the critical comparisons in the analysis of the *Trichobaris mucorea* infestation data are those made between, and not within populations, the correction of our statistics does not change the conclusions we draw from these data.

Answer to Question 2: Yes, the G-tests and Fisher’s tests are appropriate and generally the requirement of independent observations is not violated. However, one comparison shown in Figure 10 and a few comparisons in the associated results text violated the independence assumption of the Fisher’s test. We have removed these comparisons. The corrected analysis does not change our conclusions at all.